# Renewed proliferation in adult mouse cochlea and regeneration of hair cells

Yilai Shu[1,2,3,4,10], Wenyan Li[1,2,3,4,10], Mingqian Huang[1,2,10], Yi-Zhou Quan[1,2,10], Deborah Scheffer[1,2,5], Chunjie Tian[1,2], Yong Tao[1,2], Xuezhong Liu[6], Konrad Hochedlinger [7,8,9], Artur A. Indzhykulian[1,2], Zhengmin Wang[3,4], Huawei Li[3,4] & Zheng-Yi Chen [1,2]*

The adult mammalian inner ear lacks the capacity to divide or regenerate. Damage to inner ear generally leads to permanent hearing loss in humans. Here, we present that reprogramming of the adult inner ear induces renewed proliferation and regeneration of inner ear cell types. Co-activation of cell cycle activator *Myc* and inner ear progenitor gene *Notch1* induces robust proliferation of diverse adult cochlear sensory epithelial cell types. Transient MYC and NOTCH activities enable adult supporting cells to respond to transcription factor *Atoh1* and efficiently transdifferentiate into hair cell-like cells. Furthermore, we uncover that mTOR pathway participates in MYC/NOTCH-mediated proliferation and regeneration. These regenerated hair cell-like cells take up the styryl dye FM1-43 and are likely to form connections with adult spiral ganglion neurons, supporting that *Myc* and *Notch1* co-activation is sufficient to reprogram fully mature supporting cells to proliferate and regenerate hair cell-like cells in adult mammalian auditory organs.

[1] Department of Otolaryngology-Head and Neck Surgery, Graduate Program in Speech and Hearing Bioscience and Techology and Program in Neuroscience, Harvard Medical School, Boston, MA 02115, USA. [2] Eaton-Peabody Laboratory, Massachusetts Eye and Ear Infirmary, 243 Charles St., Boston, MA 02114, USA. [3] ENT Institute and Otorhinolaryngology Department of the Affiliated Eye and ENT Hospital, State Key Laboratory of Medical Neurobiology, Institutes of Biomedcial Sciences, Fudan University, 200031 Shanghai, China. [4] NHC Key Laboratory of Hearing Medicine, Fudan University, Shanghai 200031, China. [5] Department of Neurobiology and Howard Hughes Medical Institute, Harvard Medical School, Boston, MA 02115, USA. [6] Department of Otolaryngology, University of Miami School of Medicine, Miami, FL 33136, USA. [7] Department of Molecular Biology, Cancer Center and Center for Regenerative Medicine, Massachusetts General Hospital, Boston, MA 02114, USA. [8] Department of Stem Cell and Regenerative Biology and Harvard Stem Cell Institute, Cambridge, MA 02138, USA. [9] Howard Hughes Medical Institute, Chevy Chase, MD 20815, USA. [10]These authors contributed equally: Yilai Shu, Wenyan Li, Mingqian Huang, Yi-Zhou Quan *email: Zheng-Yi_Chen@meei.harvard.edu

One of the most common human sensory disorders, hearing loss affects approximately one in 500 newborns and more than half of the population over 70 years of age[1,2]. Despite its prevalence, there remains no available pharmacological therapies to treat hearing loss. Loss of hair cells (HCs), the inner ear sensory cells that detect sound and sense balance, is a major cause of hearing loss and vestibular dysfunction in humans. In lower vertebrates such as birds, fish, and amphibians, HC loss triggers supporting cells (SCs) to re-enter the cell cycle[1–4]. Proliferating SCs then transdifferentiate into new HCs, resulting in the recovery of hearing and vestibular functions[5]. In contrast, the mature mammalian cochlea completely lacks the capacity to spontaneously proliferate or regenerate HCs, and has very limited regeneration potential in the vestibular system[6,7].

To recover from deafness due to HC loss, reprogramming of adult SCs is likely necessary for renewed proliferation and transdifferentiation of SCs to HCs (SC-to-HC). In lower vertebrates, these two events occur spontaneously[1,2,8], however, the underlying mechanisms remain largely unknown. In culture, neonatal mouse inner ear SCs retain the capacity to proliferate and transdifferentiate into HCs[9], yet, this capacity is lost in the adult inner ear. As SCs serve essential roles in hearing and balance and deficits in SCs can result in deafness[10], an ideal approach to regenerate HCs by SC transdifferentiation would require replacement of lost SCs. Thus, renewed proliferation is likely best suited to achieve a balance between HC regeneration and maintenance of the SC population.

Multiple cell cycle genes, including *Rb1*, *Cdkn1b* (p27Kip1), *Cdkn2d* (p19Ink4d), and *Cdkn1a* (p21Cip1)[11–16], have been studied in induction of proliferation in the mammalian inner ear, however, none were sufficient in inducing proliferation in the adult cochlea. In the young mammalian inner ear, SC-to-HC transdifferentiation can be induced by overexpression of HC fate-determining transcription factor, *Atoh1*[17]. An early study provided evidence that *Atoh1* overexpression had limited but similar effects in the adult mammalian cochlea, however, subsequent studies failed to reproduce the essential findings[18–22]. It is therefore suggested that, in the adult inner ear, overexpression of *Atoh1* in SCs alone is inefficient in promoting HC regeneration. To recapture the capacity to respond to HC induction signals, it is likely that mature SCs need to first regain the properties of their younger biological selves.

To identify potential reprogramming factors in the adult mammalian inner ear, we began by studying chick and zebrafish HC regeneration models and uncovered that reactivation of *Myc* is a major event that leads to cell cycle re-entry[23], suggesting that a similar mechanism could induce proliferation in the mammalian inner ear. Additional studies have shown that overexpression of *Notch1*, a receptor important in mammalian inner ear early development and patterning, is sufficient to induce formation of the prosensory domain of the developing mouse otocyst;[24,25] indicating the prominent role of *Notch1* in conferring prosensory domain properties. We hypothesize that the combined action of MYC and NOTCH1 may be sufficient to reprogram adult mouse inner ear cells for cell cycle re-entry and the reprogrammed SCs may regain the properties enabling them to transdifferentiate into HCs in the presence of induction signals.

In this study, by adenovirus-mediated delivery and inducible transgenic mouse models, we demonstrate the proliferation of both HCs and SCs by combined *Notch1* and *Myc* activation in in vitro and in vivo inner ear adult mouse models. These proliferating mature SCs and HCs maintain their respective identities. Moreover, when presented with HC induction signals, reprogrammed adult SCs transdifferentiate into HC-like cells both in vitro and in vivo. We identify the mTOR pathway as downstream of *Myc/Notch1* activation and therefore a required player in proliferation and SC-to-HC transdifferentiation in the adult cochlea. Finally, our data suggest that regenerated HC-like cells likely possess functional transduction channels and are able to form connections with adult auditory neurons.

## Results

**_Myc/Notch1_ co-activation induces division in adult inner ear**. In lower vertebrates, SC proliferation and transdifferentiation are major mechanisms involved in HC regeneration[8]. In zebrafish model after HC damage, reactivation of *c-myc* (*myc*) is necessary for renewed SC proliferation[23]. Further Notch is activated during zebrafish HC regeneration[26]. To evaluate the direct role of *Myc* in renewed proliferation in the mouse inner ear, we used the cochleostomy technique to inject adenovirus carrying human *MYC* (ad-*Myc*)[27] into the mouse cochlear scala media, and studied proliferation via BrdU incorporation. BrdU was detected in postnatal day 7 (P7) cochlear SCs (SOX2[+]) (Supplementary Fig. 1a–d), but not in the adult mouse cochlea (Supplementary Fig. 1f). The cochleostomy procedure in adults damages a majority of the outer hair cells (OHCs) around the injection site, consistent with our previous study[28]. To study the independent effect of *Notch1* activation, we injected an adenovirus carrying *Cre* recombinase gene (ad-*Cre*) into the cochlea of adult Rosa-NICD transgenic mice[29] which carry a floxed stop codon that blocks expression of the *Notch1* intracellular domain (NICD). NICD activation alone did not induce proliferation (Supplementary Fig. 1g).

We hypothesized that reprogramming by combined action of inner ear progenitor genes and cell cycle activators is necessary to induce proliferation in adult cochlea. We determined the combined effect of *Myc* and *Notch1* co-activation by injecting a mixture of ad-*Myc*/ad-*Cre* virus into fully mature (6 weeks) Rosa-NICD cochlea, followed by BrdU intraperitoneal (i.p.) injection in vivo (Fig. 1a). Checking at two different time points, four and 35 days after injection, we found proliferating inner hair cells (IHCs) (MYO7A[+]/BrdU[+]) and SCs (SOX2[+]/BrdU[+]) at the injection site in the injected cochlea (Fig. 1b–i and n–o). In comparison, no proliferating cells were found in the ad-V5-injected control adult Rosa-NICD cochlea (Fig. 1j–o; Supplementary Fig. 1j) or in the uninjected cochlea (Supplementary Fig. 1h).

Regeneration studies in the adult mammalian inner ear are particularly challenging due to the lack of an efficient in vitro system that allows such studies. To overcome this limitation and effectively study reprogramming and proliferation in adult cochleae, we established a system of adult whole cochlea explant culture (Supplementary Fig. 2). This was achieved by culturing adult cochlea encased in bone with the opening in the apex and the base (see Methods for details). In this model, while all OHCs and a majority of IHCs die rapidly in culture, a majority of SCs (SOX2[+]) survive for over two weeks while maintaining an intact overall cochlear structure (Supplementary Fig. 2).

Cell division is marked by a series of distinct cellular events. To study cell cycle, we cultured adult Rosa-NICD cochleae and infected them with ad-*Myc*/ad-*Cre* (Fig. 2a). We identified SCs in mitosis by mitotic figures and MKI67 labeling (Fig. 2b–d, h). Dividing SCs were labeled with mitosis (phosphorylated histone H3, pH3) (Fig. 2e, i) and cytokinesis (Aurora B, AURKB; α-tubulin, TUBA4A) (Fig. 2f, g) markers. We quantified the number of SCs (SOX2[+]) seven days after ad-*Myc*/ad-*Cre* infection in Rosa-NICD cochleae in the mid-apical region, which included outer pillar cells, Deiters' cells, Hensen cells and Claudius' cells (Fig. 1a), and found a significant increase in the

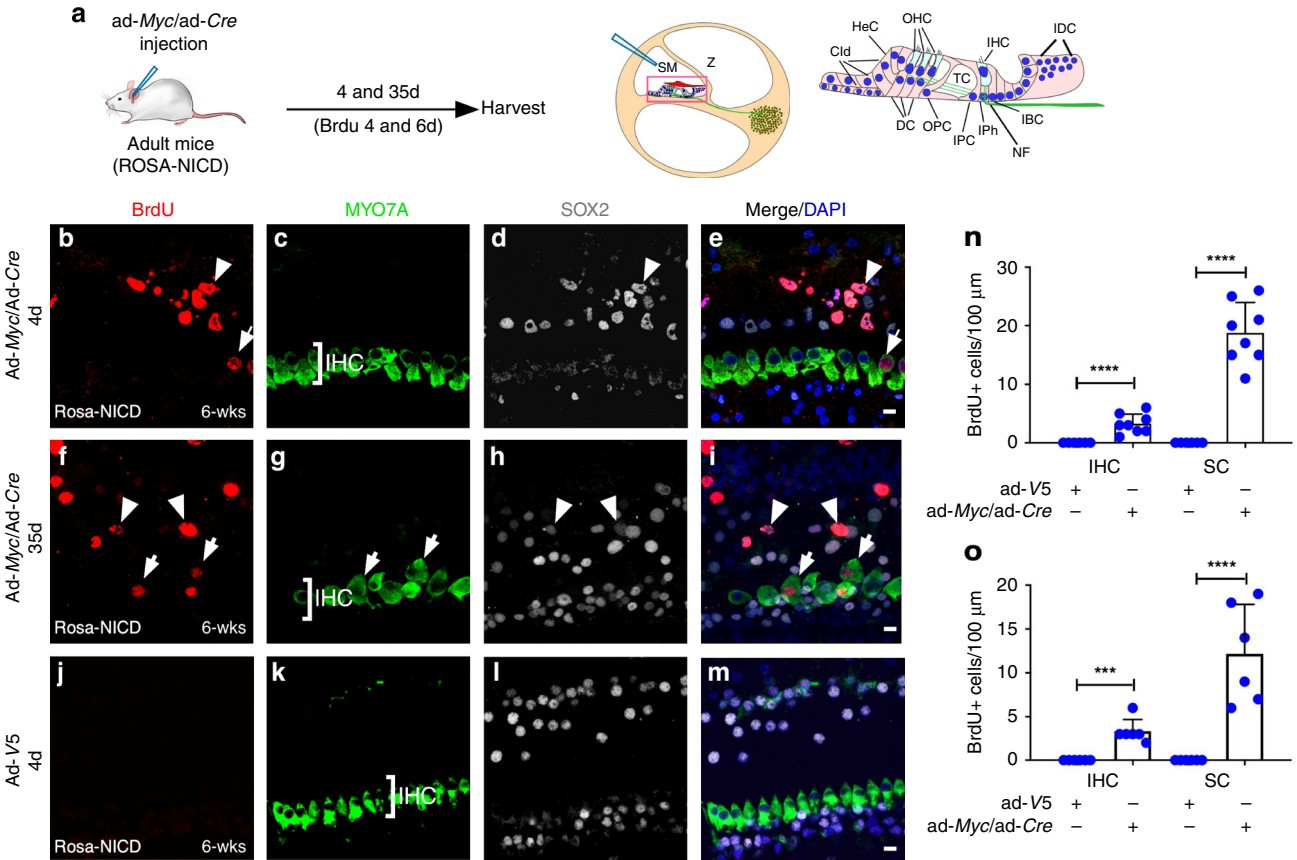

**Fig. 1** *Myc* and *NICD* co-activation induces proliferation in adult mouse cochlea in vivo. **a** A diagram illustrating the procedure of ad-*Myc*/ad-*Cre* injection in adult Rosa-NICD cochlea (left). A diagram depicts injection into the scala media (SM) of adult cochlea by cochleostomy (middle). Enlarged inset of a cross section shows cochlear structure and cell subtypes (right). Cld: Claudius' cells; HeC: Hensen cells; OHC: outer hair cells; IHC: inner hair cells; IDC: interdental cells; DC; Deiters' cells; OPC: outer pillar cells; TC: tunnel of Corti; IPC: inner pillar cells; IPh: inner phalangeal cells; NF: neuro fibers of auditory neurons; and IBC: inner cells. **b–i** Four (**b–e**) and 35 (**f–i**) days after injecting ad-*Myc*/ad-*Cre* mixture to the adult (6-week-old) Rosa-NICD cochleae in vivo, proliferating IHCs (MYO7A$^+$/BrdU$^+$, arrows) and SCs (SOX2$^+$/BrdU$^+$, arrowheads) were detected. **j–m** Four days after control ad-*V5* in vivo injection into the adult Rosa-NICD cochleae, no proliferating cell was detected. **n**, **o** Quantification and comparison of number and percentage of BrdU$^+$ IHCs and SCs in adult Rosa-NICD cochleae infected with ad-*Myc*/ad-*Cre* in vivo for four days (**n**) or 35 days (**o**) in the mid-base turn, compared to the ad-*V5*-injected adult Rosa-NICD cochleae. ***$p < 0.001$, ****$p < 0.0001$, two-tailed unpaired Student's *t*-test. Error bar, mean ± s.d.; $n = 8$ for ad-*Myc*/ad-*Cre* groups, $n = 6$ for control groups in (**n**); $n = 6$ for each group in (**o**). *n* is the number of biologically independent samples. Scale bars: 10 μm. Source data are provided as a Source Data file.

total number of BrdU$^+$ cells and SCs in ad-*Myc*/ad-*Cre*-infected cochleae compared to the ad-V5-infected controls (Fig. 2j, k).

To characterize IHC proliferation in response to *Myc*/*NICD* co-activation, we injected ad-*Myc*/ad-*Cre* into adult Rosa-NICD mouse cochlea in vivo via cochleostomy. Some IHCs were induced to divide (BrdU$^+$ doublets) with mitotic figures (labeled by MKI67 and pH3) (Fig. 2l–n, q–s). Thirty-five days after injection, stereocilia in proliferating IHCs were still present (Supplementary Fig. 1i, arrow). Injected adult cochleae showed dividing IHCs with enlarged cell bodies (Fig. 2o, p), consistent with *Myc* overexpression shown to increase cell size[30]. We counted the number of IHCs and found ad-*Myc*/ad-*Cre*-injected adult ROSA-NICD cochlea had more IHCs compared to the ad-V5-injected control cochleae at the injection site (Fig. 2t).

To determine whether proliferation is the direct result of co-activation of *Myc* and *NICD*, we infected cultured adult wild type cochlea explants with a mixture of ad-*Myc* and adenovirus carrying a V5-epitope-tagged *NICD* (ad-*NICD*-V5)[31] (Supplementary Fig. 3a). We found that a majority of BrdU$^+$ cells were double positive for MYC and NICD (V5) (89 ± 5%, $n = 4$ from biologically independent samples; mean ± s.e.m.), with the remaining being MYC$^+$ only (5.6 ± 3.1%) or NICD$^+$ only (3.1 ± 0.8%), or negative

for both (2.1 ± 1.4%) (Supplementary Fig. 3b). Semi-quantitative RT-PCR showed an up-regulation of *Notch1*/NICD and *Notch1* target *Hes5* after ad-*Myc*/ad-*Cre* infection of the cultured adult Rosa-NICD cochlea (Supplementary Fig. 3d). Together, this provides sufficient evidence that *Myc*/*Notch1* co-activation induces adult cochlear SCs and IHCs to re-enter and complete the cell cycle, resulting in increased SC and IHC numbers.

To study the origin of proliferating cells in adult cochlea, we performed lineage tracing using inducible *Sox2*-promoter-driven *Cre* mice (Sox2-CreER)[32] crossed with tdTomato (tdT) reporter mice. In resulting Sox2-CreER/tdT$^{f/f}$ mice, exposure to tamoxifen (Tm) induces tdT expression permanently and specifically in SCs[33]. To better understand the origin of proliferating HCs, we injected cochleae of adult Sox2-CreER/tdT$^{f/f}$ mice with either ad-*V5* or an ad-*Myc*/ad-*NICD* mixture after administering the Tm by i.p. injection (Fig. 3a). In the control ad-*V5*-injected Sox2-CreER/tdT$^{f/f}$ cochleae, SCs (SOX2$^+$) and IHCs (MYO7A$^+$) were tdT$^+$ and tdT$^-$, respectively (Fig. 3b). In the ad-*Myc*/ad-*NICD*-injected cochleae, dividing HCs were detected and all were found to be tdT$^-$ (Fig. 3c), demonstrating that they were derived from existing IHCs. To study the origin of proliferating SCs, we infected cultured adult Sox2-CreER/tdT$^{f/f}$ cochlea in vitro with ad-*Myc*/ad-*NICD* after

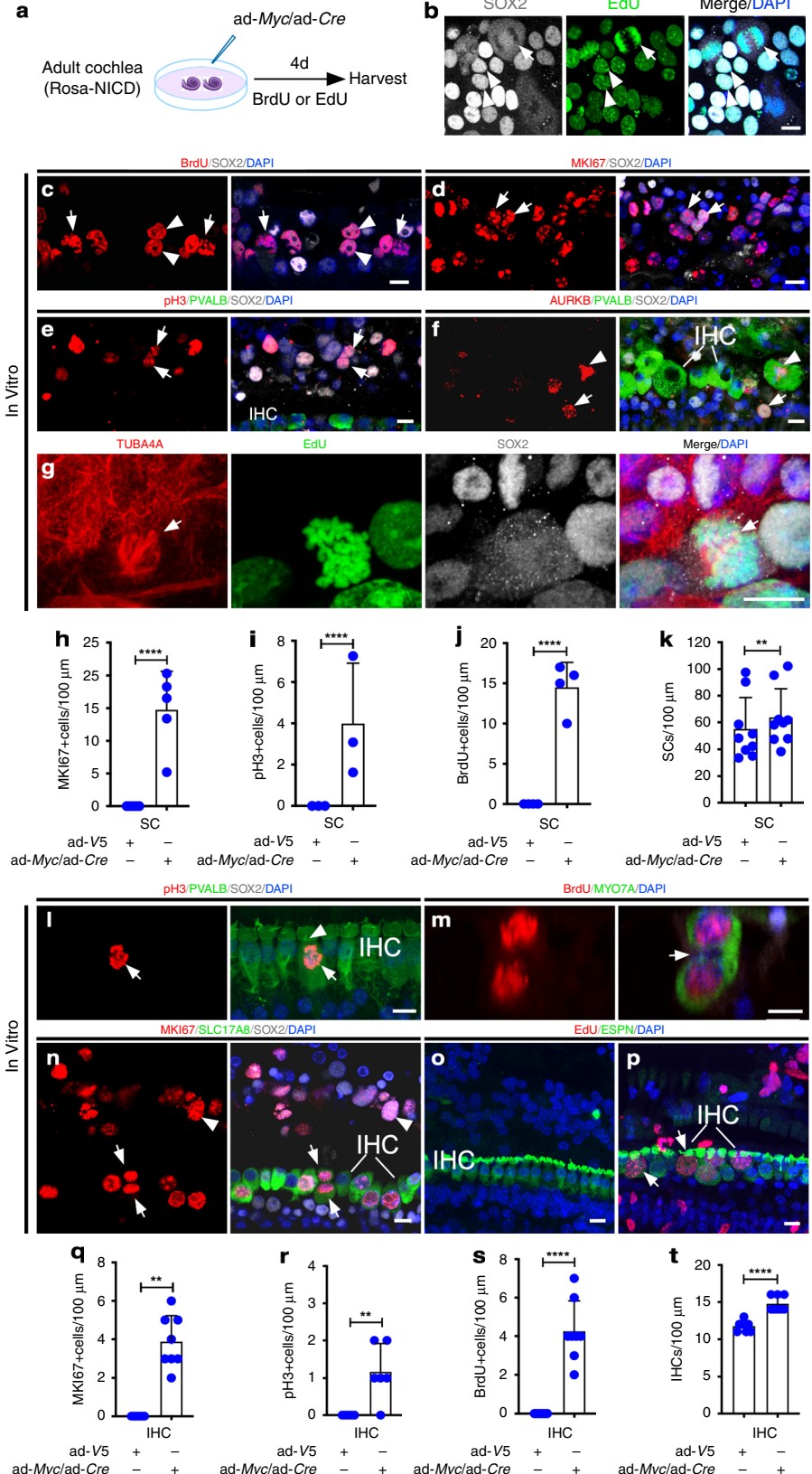

Tm treatment (Fig. 3d). We found tdT⁺ SCs co-labeled with EdU and MKI67 (Fig. 3e–j), demonstrating proliferating SCs were of SC origin. In cochleae with proliferating SCs (Fig. 3k), the overall cochlear structure was preserved as shown by the tunnel of Corti (TC, Fig. 3l–m). By the anatomic locations and by S100A1 and JAG1 labeling, we found that dividing SC subtypes are likely to include pillar cells, Deiters' cells, Hensen cells and Claudius' cells (arrows and arrowheads in Fig. 3g, l, m; Supplementary Fig. 4b, e). No EdU⁺ cell was observed in ad-V5-injected control groups (Supplementary Fig. 4c, f).

**Fig. 2** Characterization of proliferation in adult cochlea. **a** A diagram illustrating the experimental procedure. **b–f** Four days post ad-*Myc*/ad-*Cre* infection of the Rosa-NICD adult (6-week-old) cochlear explant in vitro, SC mitotic figures (**b**, **c**, arrows), dividing SC nuclear doublet (**b**, **c**, arrowheads), MKI67+ (Ki67+) SCs (**d**, arrows) were seen. pH3+ SCs (**e**, arrows), AURBK+ SCs (**f**, arrow) and AURBK+ IHCs (**f**, arrowhead) were detected. **g** Alpha-tubulin labeled dividing SCs (TUBA4A+/ SOX2+/EdU+) in vitro, with the arrow showing TUBA4A+ spindle microtubules. **h–k** Quantification showed an increase in the SCs labeled with MKI67 (**h**), pH3 (**i**) and BrdU (**j**), as well as the total number of SCs (**k**) in ad-*Myc*/ad-*Cre*-infected compared to control ad-*V5*-infected Rosa-NICD cochleae. **p < 0.01, ****p < 0.0001, two-tailed unpaired (**h–j**) and paired (**k**) Student's t-tests. Error bar, mean ± s.d.; **h**: n = 5; **i**: n = 4; **j**: n = 3; k: n = 9. **l–n** Four (**l**) and six (**m**, **n**) days after ad-*Myc*/ad-*Cre* injection into adult Rosa-NICD mouse cochlea in vivo, a pH3+ IHC (**l**, arrow; arrowhead points to hair bundles), a dividing IHC (**m**) with the arrow pointing to the cleavage site between two daughter cells, and MKI67+ IHCs (**n**, SLC17A8+/MKI67+, arrows point to a pair of IHC doublets) and SCs (**n**, SOX2+/MKI67+, arrowhead) were detected. **o**, **p** Compared to a control ad-*V5*-injected Rosa-NICD adult cochlea without proliferation (**o**), the ad-*Myc*/ad-*Cre* injected Rosa-NICD cochlea showed IHC proliferation with enlarged cell bodies (**p**, arrows show EdU+ IHC with enlarged bodies), 12 days after injection. **q**, **r** Quantification showed increase in MKI67+ IHCs (**q**) and pH3+ IHCs (**r**) in adult Rosa-NICD cochleae injected with ad-*Myc*/ad-*Cre* or ad-*V5* for 4 days, respectively. **p < 0.01, two-tailed unpaired Student's t-test. Error bar, mean ± s.d.; n = 8 for each group in (**q**); n = 6 for each group in (**r**). **s**, **t** Quantification showed increase in BrdU+ IHCs (**s**) and the total number of IHCs (**t**) in adult Rosa-NICD cochleae injected with ad-*Myc*/ad-*Cre* and ad-*V5* in vivo at six days. ****p < 0.0001, two-tailed unpaired Student's t-test. Error bar, mean ± s.d.; n = 8 for each group in (**s**, **t**). n is the number of biologically independent samples. Scale bars: 10 μm. Source data are provided as a Source Data file.

To assess possible cell death after viral infection, we performed cleaved-caspase-3 (CASP3) labeling using cultured adult Rosa-NICD cochleae infected by ad-*Myc*/ad-*Cre*, as infection rate was more uniform in culture. Compared to ad-*GFP*-infected control cochleae, ad-*Myc*/ad-*Cre*-infected cochleae showed significantly more CASP3+ cells in the mid-apex in the sensory epithelial region (Supplementary Fig. 5a–f), supporting that MYC/NICD activation induces apoptosis. While cell death is consistent with high and sustained MYC activity due to adenovirus-mediated overexpression, our subsequent experiment showed that under transient *Myc/Notch* activation, proliferating cells can survive.

**Production of HC-like cells by transient *Myc/NICD* activation.** In contrast to lower vertebrates, in which HC loss results in SCs dividing and spontaneously transdifferentiating into HCs[1–4], after *Myc/NICD* induced SC proliferation in vivo, we did not detect new HCs. There were no new HCs in the OHC region where the surgical procedure by cochleostomy killed all OHCs[28] (Fig. 1b–i) or SC-derived HCs in the lineage-tracing study (Fig. 3), indicating the lack of spontaneous transdifferentiation from SCs to HCs upon cell cycle re-entry. Despite the evidence that overexpression of *Atoh1* induces cochlear SCs to transdifferentiate into HCs in neonatal and embryonic mice[17,34], such transdifferentiation by *Atoh1* overexpression is extremely inefficient at the adult stage[21]. To determine if *Atoh1* promotes dividing adult SCs to transdifferentiate to HCs, we performed a triple infection with mixed ad-*Myc*/ad-*Cre*/ad-*Atoh1* in the cultured adult Rosa-NICD cochlea and analyzed HC regeneration. To our surprise, no new HCs were generated in cultured cochlea after triple infection (Supplementary Fig. 6).

The lack of SC-to-HC transdifferentiation, either spontaneously or in response to *Atoh1* overexpression, could be due to sustained *Myc/NICD* co-activation that promotes proliferation but prevents transdifferentiation[35]. To promote SC-to-HC transdifferentiation, it may be necessary to transiently activate *Myc/NICD* to reprogram SCs and then follow with HC induction signals.

To activate *Myc/Notch1* transiently, we created a doxycycline (Dox)-inducible transgenic mouse line by crossing Rosa26-promoter-driven reverse tetracycline-controlled transactivator (rtTA) transgenic mice with tetracycline-response-element-controlled *Myc* and *NICD* mice (tet-*Myc* and tet-*NICD*), creating an rtTA/tet-*Myc*/tet-*NICD* inducible model. Using explant culture, we showed that *Myc* and *NICD* expression are tightly and reversibly controlled by the exposure and withdrawal of Dox and that renewed proliferation is the direct result of *Myc/NICD* co-activation (Supplementary Fig. 7a–e).

Adult SCs and IHCs do not change their cell identities when dividing, suggesting limited reprogramming, which can be shown by re-expression of progenitor but not embryonic stem cell genes.

By qRT-PCR of cultured adult Dox-treated rtTA/tet-*Myc*/tet-*NICD* cochleae (Supplementary Fig. 7a), we were able to detect the re-expression of inner ear progenitor genes including *Six1*, *Eya1*, *Gata3* and *Isl1* (Supplementary Fig. 7f). These genes are downstream of the Notch pathway and are involved in the formation of the auditory sensory epithelium[34,36]. Stem cell genes such as *Nanog* and *Fut4* (*Ssea1*) were not upregulated. *Cdkn1b* (*p27Kip1*), a negative cell cycle regulator in the SCs[12], was significantly down-regulated, supporting that MYC/NICD activities suppress *Cdkn1b* expression, which is, in part, involved in adult SC proliferation. The data support the idea that there is limited reprogramming by MYC/NICD-driven reactivation of inner ear progenitor genes.

To test the hypothesis that transient reprogramming is necessary for adult SC-to-HC transdifferentiation, we reversibly activated *Myc/NICD* in vivo by placing Dox-soaked gelfoam (50 mg/ml) in the round window niche of adult rtTA/tet-*Myc*/tet-*NICD* cochleae for four days followed by ad-*Atoh1* injection via cochleostomy (Fig. 4a). Two weeks after ad-*Atoh1* injection, control adult rtTA/tet-*Myc*/tet-*NICD* mice without Dox exposure had no new HCs (Fig. 4b). In Dox-treated rtTA/tet-*Myc*/tet-*NICD* cochlea followed by ad-*Atoh1* infection, we found ectopic HC-like cells (eHCs, defined as regenerated HC-like cells), labeled with HC markers ESPN and PVALB, regenerated in different cochlear areas (Fig. 4c) including the IHC region (Fig. 4d–f). Overall, the additional number of regenerated HC-like cells resulted in a cumulative increase in total HC number in comparison to the controls (Fig. 4i, j). Some of the eHCs were EdU+ indicating regenerated HC-like cells were derived from dividing SC transdifferentiation (Fig. 4f, h). A large number of ectopic HC-like cells (ESPN+) were regenerated in the limbus region (Fig. 4d, j; Supplementary Movie 1), the Claudius cell region (CldR) and the cochlear sensory epithelial region (SE) (Fig. 4i; Supplementary Fig. 8a, b). The majority of regenerated HC-like cells in the SE or CldR regions were from non-dividing (EdU−) cells (Supplementary Fig. 8a, b).

We further determined whether transient *Myc/NICD* activation is required for HC regeneration in adult mouse in vitro. Adding Dox to cultured adult rtTA/tet-*Myc*/tet-*NICD* cochleae for three days and following with ad-*Atoh1* infection lasting for 14 days (Fig. 5a), we detected no new HCs in the Dox-only-treated control cochlea (Fig. 5b). In the ad-*Atoh1*-only-infected control cochlea, a small number of new HC-like cells were detected (Fig. 5c, e, f), all without induction of proliferation. In the adult rtTA/tet-*Myc*/tet-*NICD* cochlea explant treated with Dox transiently followed by *ad-Atoh1* infection, we detected robust HC regeneration (MYO7A+/PVALB+) in the sensory epithelial region (SE), the limbus regions (Lib) (Fig. 5d), and from transdifferentiation of both

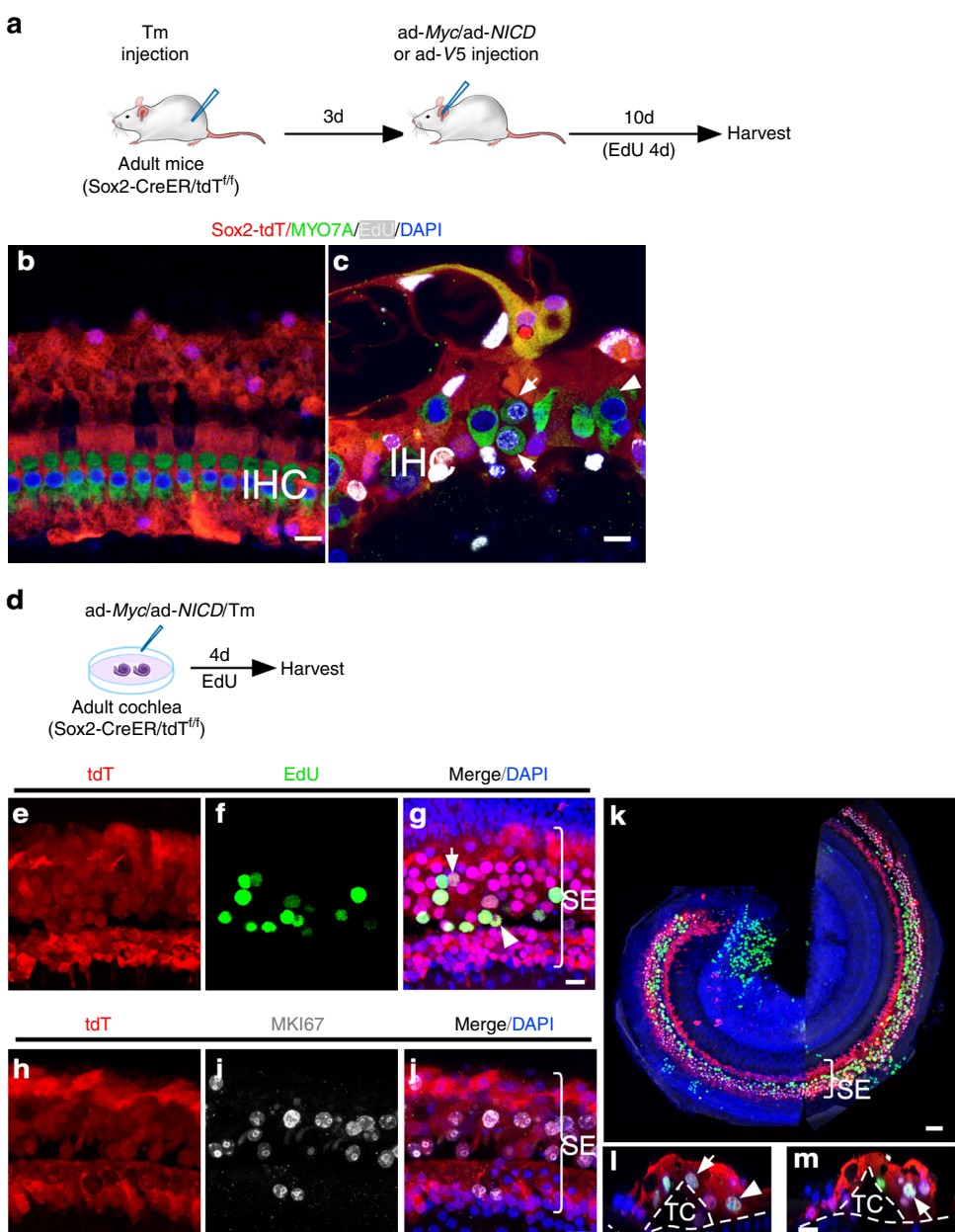

**Fig. 3** Study of the origin of dividing HCs and SCs in adult cochlea by lineage tracing. **a** A schematic diagram depicting the in vivo experimental procedure. **b** In control adult Sox2-CreER/tdT$^{f/f}$ mice 10 days after ad-$V5$ injection and Tamoxifen (Tm) treatment, all IHCs were tdT negative and were surrounded by tdT$^+$ SCs. No EdU labeling was detected. **c** Following Tm treatment and ad-$Myc$/ad-$NICD$ injection into adult Sox2-CreER/tdT$^{f/f}$ mouse cochlea for 10 days, dividing IHCs were detected (arrows). All IHCs (dividing, arrows; non-dividing, arrowhead) were tdT negative. **d** Schematic diagram of lineage-tracing of proliferating adult SCs in vitro. **e–g** EdU labeling in the SCs (tdT$^+$) in cultured adult Sox2-CreER/tdT$^{f/f}$ cochlea after treatment with Tm and infection with ad-$Myc$/ad-$NICD$. Judging by their locations, likely dividing Deiters' cells (arrow) and pillar cells (arrowhead) were identified. Bracket indicates the sensory epithelium (SE). **h–j** MKI67 labeling in the SCs (tdT$^+$) in cultured adult Sox2-CreER/tdT$^{f/f}$ cochlea after treatment with Tm and infection with ad-$Myc$/ad-$NICD$. **k** A confocal image of the surface view of the middle turn of adult Sox2-CreER/tdT$^{f/f}$ cochlea after treatment with Tm and infection by ad-$Myc$/ad-$NICD$ to show numerous proliferating SCs (tdT$^+$/EdU$^+$) in the SE region. **l, m** z-axis cross-section views of confocal images in (**k**) showed dividing SCs by triple labeling of tdT$^+$/EdU$^+$/SOX2$^+$ (likely Deiters' cells, arrows; Hensen cell, arrowhead. tdT: red; EdU: green; SOX2: white). Dashed lines mark the basilar membrane and the tunnel of Corti (TC). Scale bars: 50 μm in (**k**); 10 μm in other figures.

dividing and non-dividing SCs (MYO7A$^+$/SOX2$^+$/EdU$^+$ and MYO7A$^+$/SOX2$^+$/EdU$^-$) (Arrows and arrowheads in Fig. 5d), respectively. Significantly more HC-like cells were regenerated after transient $Myc$/NICD activation followed by $Atoh1$ induction than $Atoh1$ induction alone (Fig. 5e, f). The cochlear regions in which HC-like cells were regenerated including the sensory epithelial region, the limbus region, and the Claudius cell region, were consistent between in vivo and in vitro studies. We, therefore,

conclude that $Myc$/NICD co-activation successfully reprograms SCs, and their subsequent downregulation enables reprogrammed SCs (dividing and non-dividing) to then transdifferentiate into HC-like cells in response to $Atoh1$ for both in vivo and in vitro.

We hypothesized that sustained MYC/NICD activities leads to increase in apoptosis (Supplementary Fig. 5a–f). Conversely, transient MYC/NICD activities should lead to improved cell survival. To explore this idea, we performed study to compare cell

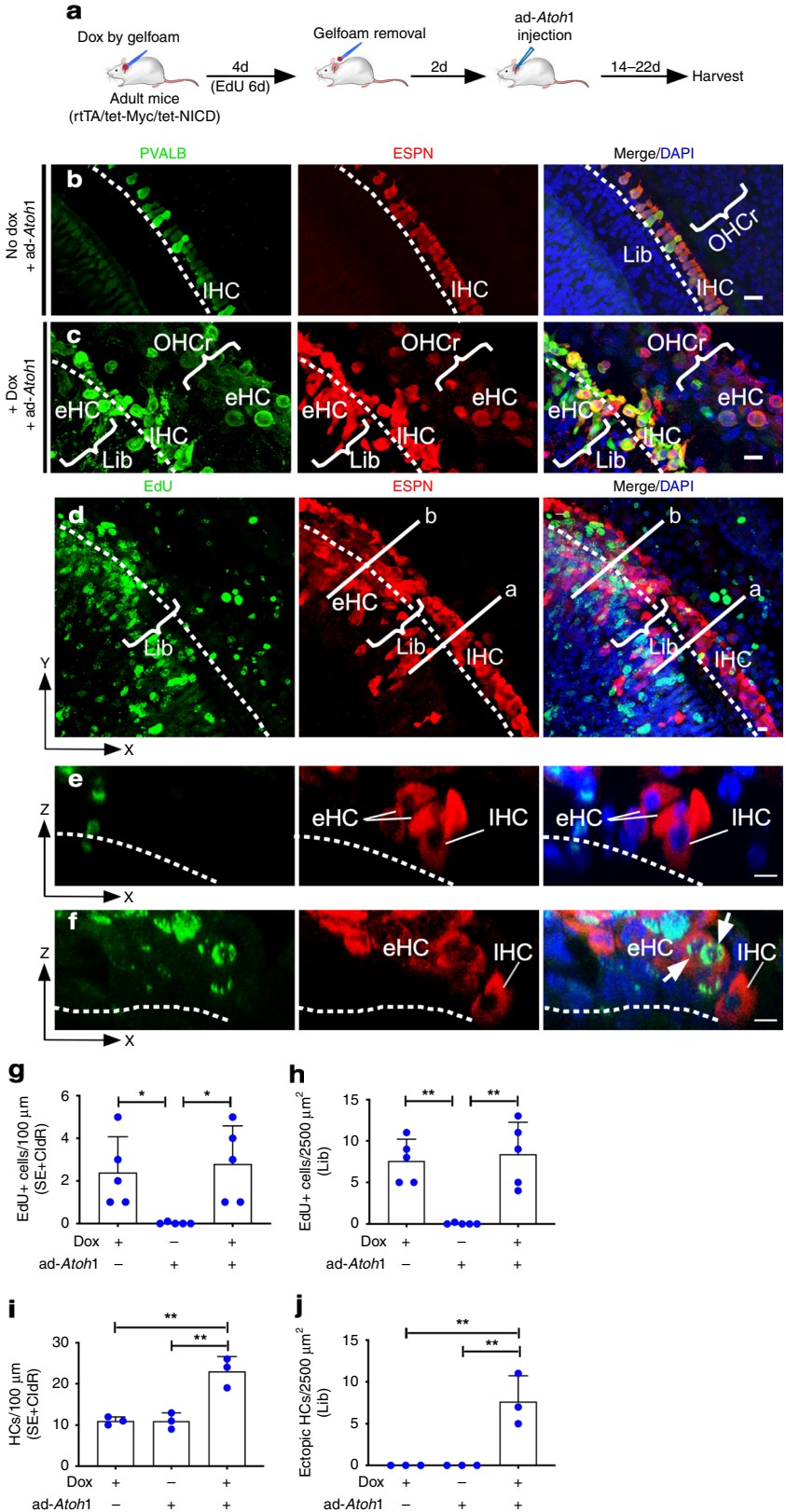

death index in cultured adult rtTA/tet-*Myc*/tet-*NICD* cochleae exposed to Dox for 7 or 14 days, respectively, with a total culture time of 14 days (Supplementary Fig. 5m). By CASP3 labeling, we found that cochleae exposed to Dox for 7 days had significantly fewer CASP3$^+$ cells in the sensory epithelial region and a majority of SCs appeared healthy (Supplementary Fig. 5g–i, n). In contrast, the cochleae exposed to Dox for 14 days had a dramatic increase in the number of CASP3$^+$ cells in the sensory epithelial region and the limbus region (Supplementary Fig. 5j–l,n). These results strongly support that transient *Myc*/*NICD* expression improves survival in reprogrammed SCs in adult cochlea, whereas continuous overexpression of *Myc*/*NICD* is detrimental to SC

**Fig. 4** Transient *Myc/NICD* activation is necessary for *Atoh1*-induced transdifferentiation of HC-like cells in vivo. **a** A schematic diagram illustrating the experimental procedure of transient Dox induction of *Myc/NICD* and HC induction by ad-*Atoh1* in vivo. **b** In control adult (P30) rtTA/tet-*Myc*/tet-*NICD* mice injected with ad-*Atoh1* by cochleostomy for 20 days in vivo, existing IHCs were co-labeled with ESPN and PVALB, whereas all OHCs died because of the procedure in the mid-base turns. No new HCs were regenerated. Dashed line demarcated the boundary between IHCs and the limbus region (Lib). Bracket indicated OHC region (OHCr). **c** In adult rtTA/tet-*Myc*/tet-NICD mice treated with Dox for 4 days followed by ad-*Atoh1* infection for 22 days, new ectopic HC-like cells (PVALB$^+$/ESPN$^+$) were detected in the OHC region (bracket, OHCr) and IHC region (eHC, bracket). **d** Upon gelfoam-mediated Dox induction to activate *Myc/NICD* for 4 days and ad-*Atoh1* infection in adult rtTA/tet-*Myc*/tet-*NICD* mice in vivo for 14 days, SCs and non-SCs re-entered cell cycle (EdU$^+$) and transdifferentiated to HC-like cells (ESPN$^+$). The existing IHCs were indicated (IHC), and the dashed line separated the IHC region from the limbus region (Lib). **e** A cross section view along the line (a) in (**d**), to illustrate an IHC and regenerated eHCs from SC transdifferentiation without proliferation. **f** A cross section view along the line (b) in (**d**), to show numerous eHCs (ESP$^+$) regenerated in the limbus region, some of which were labeled with EdU (ESPN$^+$/EdU$^+$, arrows). The dashed lines demarcated the basilar membrane in (**e**, **f**). **g-j** Quantification in the mid-base turn showed significant increases in the numbers of EdU$^+$ cells (**g-h**), total HCs in the SE + CldR (**i**) and ectopic HC-like cells in Lib regions (**j**) in the experimental groups around the injection sites after Dox treatment and ad-*Atoh1* infection, compared to the ad-*Atoh1*-injected control adult rtTA/tet-*Myc*/tet-*NICD* inner ear. *$p < 0.05$, **$p < 0.01$, one-way ANOVA with Tukey's multiple comparison test. Error bar, mean ± s.d.; (**g**, **h**): $n = 5$; (**i**, **j**): $n = 3$. $n$ is the number of biologically independent samples. Scale bars: 10 μm. Source data are provided as a Source Data file.

survival. Transient *Myc/NICD* overexpression is thus essential to the survival of reprogrammed SCs as well as their capacity to transdifferentiate into HC-like cells.

**Characteristics of new HC-like cells in adult cochlea.** Auditory HCs can be divided into two specialized groups with distinct functions: OHCs and IHCs. Cochlear OHCs can be characterized by their anatomic position, electromotility properties, and their expression of OHC marker prestin (SLC26A5)[35]. We performed SLC26A5 labeling in rtTA/tet-*Myc*/tet-*NICD* cochleae after Dox/ad-*Atoh1* treatment (Fig. 6a) and detected SLC26A5 in some regenerated HC-like cells (28 ± 13%, mean ± s.e.m., $n = 5$ biologically independent samples) in the OHC region (Fig. 6b–e), indicating that a portion of regenerated HC-like cells in the OHC region start to express OHC markers.

What are the characteristics of regenerated HC-like cells? We hypothesized that new HC-like cells transdifferentiated from adult SCs may still have characteristics resembling developmentally young HCs, while those regenerated from adult IHCs may have a more mature differentiated status given their mature HC origin. To test this hypothesis, we used acetylated tubulin (Ac-TUBA4A) labeling to examine the presence of kinocilia, a structure only associated with young but not adult auditory HCs (Supplementary Fig. 9a)[37]. We found Ac-TUBA4A$^+$ HC-like cells in the OHC region in the Dox/ad-*Atoh1*-treated rtTA/tet-*Myc*/tet-*NICD* cochlea explant culture (Fig. 6f–h). PTPRQ, a marker for young IHCs and OHCs only evident in young mice up to 14 days postnatal (Supplementary Fig. 9b)[38], was detected in regenerated HC-like cells in adult cochlea (Fig. 6j–m). In contrast, HC-like cells regenerated from the division of existing HCs did not possess kinocilia (Fig. 6i). These results provide evidence to support "young" HC status of regenerated HC-like cells from SC transdifferentiation.

The stereocilia structure of HC-like cells were detected by actin and ESPN labeling (Fig. 6n–q). Using scanning electron microscopy (SEM), we identified HC-like cells by the presence of kinocilia and immature concentric stereocilia. In normal cochlea, OHC stereocilia are organized into a recognizable V-formation (arrow, Fig. 6u) while IHC stereocilia are aligned across the cell's apical surface (arrowhead, Fig. 6u; Supplementary Fig. 10i). In contrast, immature stereocilia in regenerated HC-like cells are consistently short, either uniform in height, or disorganized (Fig. 6n–q, r–t; Supplementary Fig. 10a–h). In the control adult rtTA/tet-*Myc*/tet-*NICD* cochlea explant culture treated with Dox only or infected by ad-*Atoh1* alone, no new HC-like cells were detected by SEM (Supplementary Fig. 10j–l). Regenerated HC-like cells were able to survive in culture for two weeks, reminiscent of neonatal auditory HCs that can survive in culture for a similar timeframe. In contrast, adult cochlear HCs usually die within 24 h in culture.

To study if regenerated HC-like cells in adult cochlea have functional transduction channels, we performed an FM1-43 uptake experiment. This fluorescent dye, and its fixable analog FM1-43FX, pass through functional transduction channels and are trapped by their charge within HCs[39]. Regenerated HC-like cells in vitro, from dividing and non-dividing SCs 14 days post infection, were able to take up FM1-43FX, whereas no such uptake was detected in the surrounding cells (Fig. 6v–y).

It is essential that regenerated HC-like cells are able to make connections with surrounding auditory neurons in order to form synapses that enable hearing. We used antibodies to the neurite marker neurofilament (NEFH) to stain adult rtTA/tet-*Myc*/tet-*NICD* cochlea with regenerated HC-like cells after treatment with Dox/ad-*Atoh1* in vivo. We found extensive neurite outgrowth to the sensory epithelial region, with the neurites wrapping around new HC-like cells. In contrast, in control cochleae without HC regeneration, virtually all the neurites retracted with few in contact with the existing IHCs (Supplementary Fig. 11a–e; Supplementary Movie 2).

**mTOR is involved in *Myc/NICD* reprogramming in adult cochlea.** Understanding the MYC/NICD-mediated pathway will shed light on the reprogramming mechanism and provide opportunities to fine-tune the process for improved regeneration efficiency. MYC and NOTCH have been shown to interact with the mTOR signaling pathway[40–42]. We hypothesized that the reprogramming effect in adult cochlea by MYC/NICD may be mediated in part by the mTOR pathway. To test the hypothesis, we examined production of phospho-rpS6 (p-rpS6), an mTOR effector, in Dox-treated adult rtTA/tet-*Myc*/tet-*NICD* cochlea in culture (Fig. 7a). We detected prominent labeling of p-rpS6 in dividing SCs (SOX2$^+$/EdU$^+$/p-rpS6$^+$, arrows, Fig. 7c), but not in freshly dissected adult rtTA/tet-*Myc*/tet-*NICD* cochlea (Fig. 7b). To determine the role of mTOR in MYC/NOTCH-mediated proliferation, we incubated Dox-treated adult rtTA/tet-*Myc*/tet-*NICD* cochlea culture with rapamycin, a well-known mTOR inhibitor[43]. The p-rpS6 signal was virtually eliminated by rapamycin despite Dox treatment (Fig. 7d, e). Moreover, rapamycin treatment significantly reduced the number of EdU$^+$ cells induced by *Myc/NICD* co-activation (Fig. 7d, f). The data strongly support that mTOR is downstream of MYC/NICD and is required in MYC/NICD-mediated proliferation.

To assess if mTOR plays a role in HC regeneration in adult cochlea, we compared the number of regenerated HC-like cells in Dox-treated adult rtTA/tet-*Myc*/tet-*NICD* cochleae infected by ad-*Atoh1*, with or without rapamycin treatment (Fig. 7g). Rapamycin treatment significantly reduced the number of regenerated HC-like cells in the sensory epithelium (SE) or the

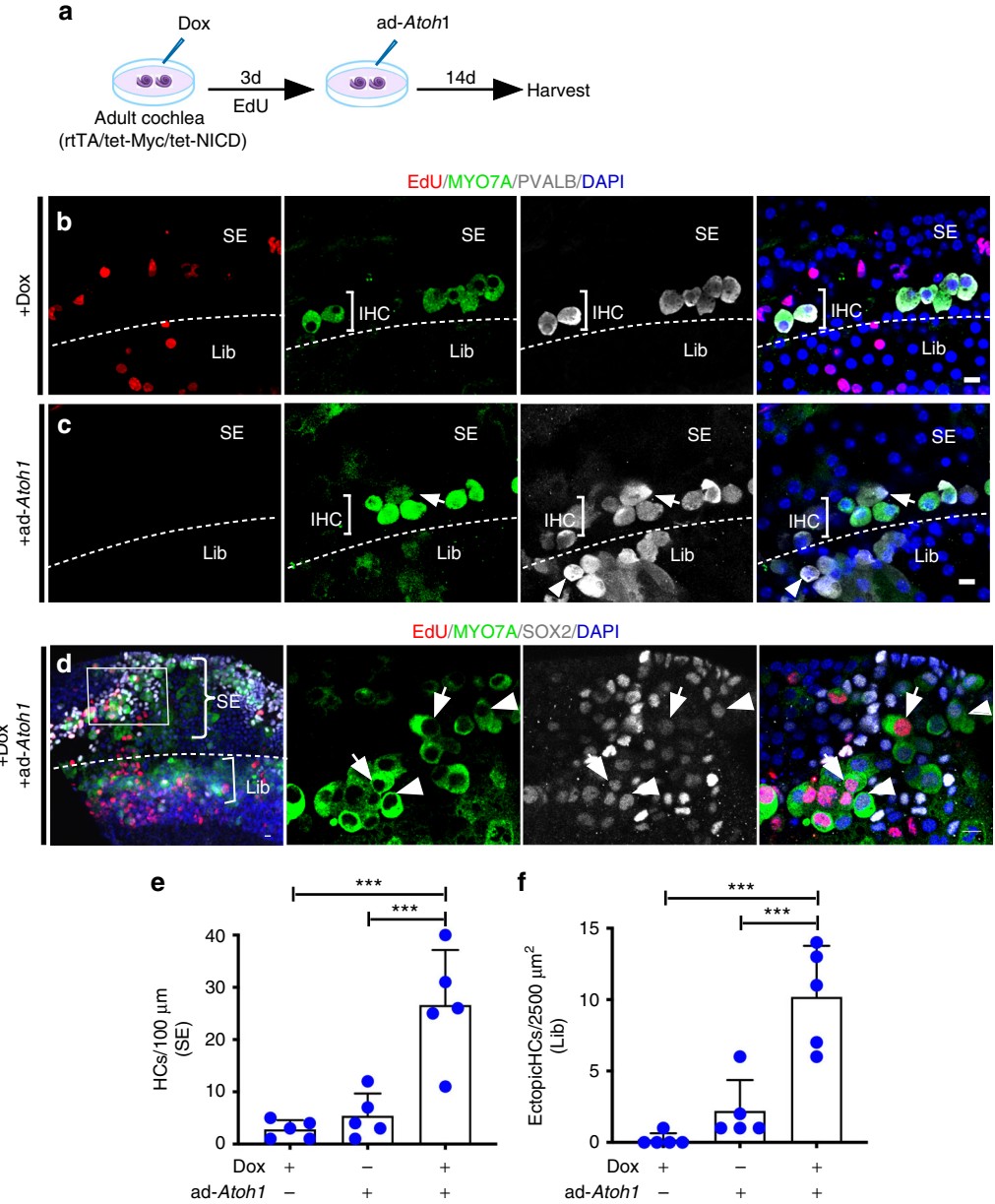

**Fig. 5** *Atoh1*-induced regeneration of HC-like cells through transient activation of *Myc/NICD* in vitro. **a** A schematic diagram illustrating in vitro transient Dox induction of *Myc/NICD* and HC induction by ad-*Atoh1* in adult rtTA/tet-*Myc*/tet-NICD cochlea explant culture. **b** No new HCs were detected in the OHC region of the sensory epithelium (SE) in the cultured adult rtTA/tet-*Myc*/tet-*NICD* cochlea following 3 days in Dox and 14 days without Dox. Proliferating cells (EdU$^+$) and some surviving IHCs (MYO7A$^+$/PVALB$^+$) were seen. **c** In the adult rtTA/tet-*Myc*/tet-*NICD* cochlea infected with *ad-Atoh1* for 14 days, PVALB$^+$ cells in the limbus region (Lib) were detected. Weak MYO7A was detected in some PVALB$^+$ cells, with arrow and arrowhead pointing to such cells in the SE and Lib regions, respectively. **d** The mid-apex of the adult rtTA/tet-*Myc*/tet-*NICD* cochlea treated with Dox for 3 days followed by ad-*Atoh1* transduction for 14 days. Left, a low-magnification image showed whole-mount adult rtTA/tet-*Myc*/tet-*NICD* cochlea treated by Dox and ad-*Atoh1* with new HC-like cells from the sensory epithelium (SE, inset) and the limbus region (Lib). Middle to right, the enlarged inset showed new HC-like cells from dividing (MYO7A$^+$/EdU$^+$, arrows) and non-dividing (MYO7A$^+$/EdU$^-$, arrowheads) SCs, all with SOX2 expression, indicating their SC origin. The dashed line demarcated the boundary between the SE and Lib regions in (**b**–**d**). **e**, **f** Quantification and comparison of regenerated HC-like cells (MYO7A$^+$) in the apical turn of the cochlea culture. ***$p < 0.001$, one-way ANOVA with Tukey's multiple comparison test. Error bar, mean ± s.d.; $n = 5$ biologically independent samples for each group. Scale bars: Source data are provided as a Source Data file.

limbus regions (Fig. 7h–j), supporting that mTOR is further required for HC regeneration.

Given the prominent role that mTOR plays in proliferation and cell growth, we hypothesized that this pathway may compensate for part of MYC's functionality in adult inner ear reprogramming. To determine the extent by which mTOR activation is sufficient to replace MYC and induce proliferation in adult cochlea, we activated mTOR in Dox-treated rtTA/tet-

*NICD* adult cochlea culture by MHY1485, a small molecule mTOR activator[44] (Fig. 7k). We observed EdU$^+$ cells in the SE and the limbus regions after MHY1485 treatment, but not in control rtTA/tet-*NICD* cochlea exposed to Dox alone (Fig. 7l–n). The number of EdU$^+$ cells was fewer after MHY1485 treatment and NICD activation compared to *Myc/NICD* co-activation (Fig. 7c), indicating that mTOR partially compensates for MYC function in renewed proliferation. We also observed more

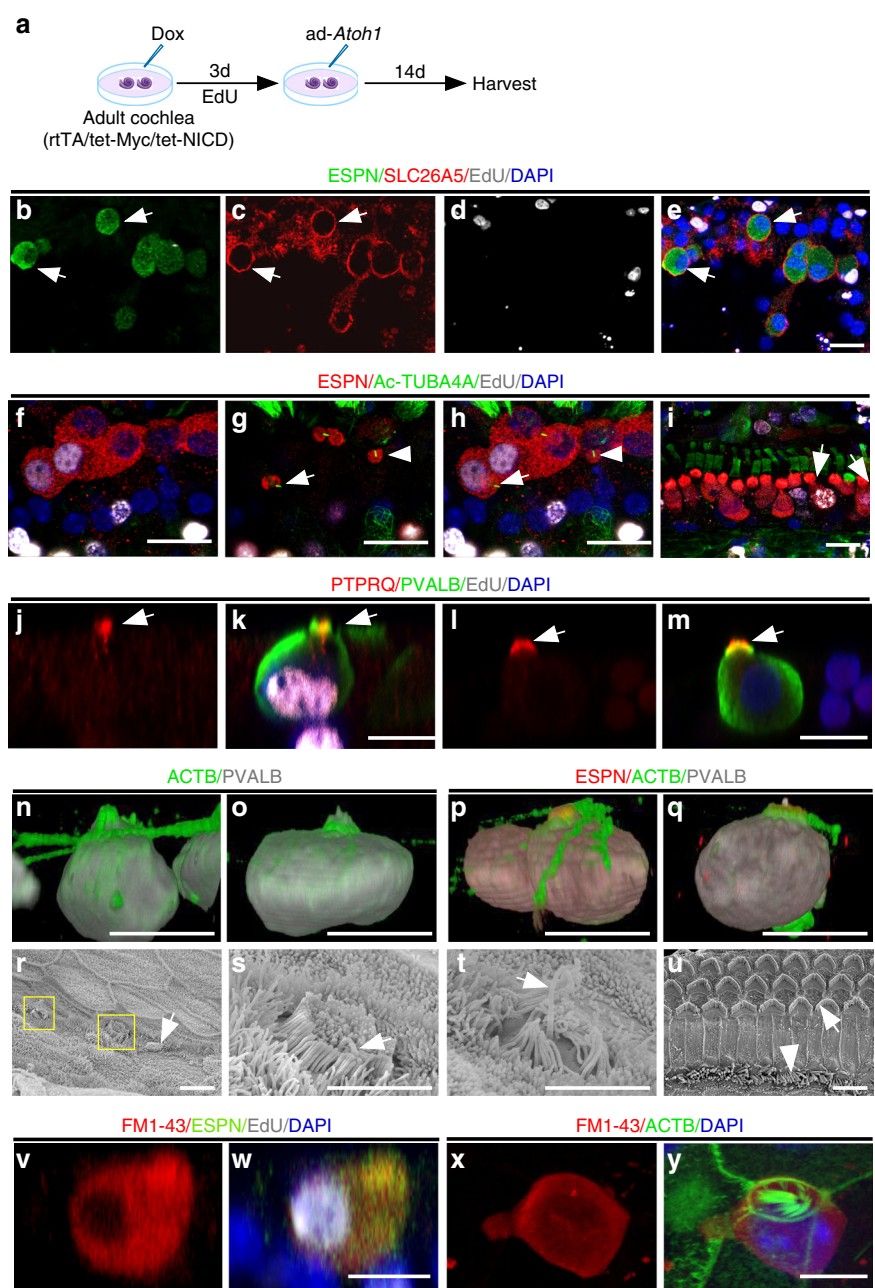

**Fig. 6** Characteristics of regenerated HC-like cells. **a** A diagram illustrating the procedure of Dox and ad-*Atoh1* treatment. **b–e** Membrane staining of SLC26A5 (prestin) was detected prominently in some regenerated HC-like cells from SC transdifferentiation in the OHC region in vitro (arrows). **f–i** Kinocilia labeled with acetylated tubulin (Ac-TUBA4A) in rtTA/tet-*Myc*/tet-*NICD* cochlea treated with Dox and ad-*Atoh1*. The confocal images were from two different planes of the same sample (**f** to show the nuclei; **g** to show the apical surface) with the merged image in (**h**). Kinocilia structure was detected in HC-like cells transdifferentiated from dividing (ESPN[+]/EdU[+], arrow in **g**, **h**) and non-dividing (ESPN[+]/EdU[−], arrowhead in **g**, **h**) SCs, respectively, but not detected in the dividing IHCs (ESPN[+]/EdU[+], arrows in **i**). **j–m** 17 days after Dox/ad-*Atoh1* treatment of the cultured adult rtTA/tet-*Myc*/tet-*NICD* cochlea, PTPRQ (young OHC marker) was detected in the hair bundles of dividing (**j**, **k**, arrow) and non-dividing (**l**, **m**, arrow) HC-like cells. **n–q** 3D-reconstruction of regenerated HC-like cells (ESPN[+]/PVALB[+]) in the OHC region. Different shapes of hair bundles can be seen (labeled with phalloidin, green). **r** Images of scanning electron microscopy (SEM) showing immature stereocilia from new HC-like cells in rtTA/tet-*Myc*/tet-*NICD* cochlea treated with Dox and ad-*Atoh1* in vitro (squares and arrow). **s**, **t** Enlarged images from squares in 6r to show short and uniform height stereocilia with a kinocilium (arrow in **s**) and disorganized stereocilia with a kinocilium (arrow in **t**). **u** An image of SEM of adult HC stereocilia in freshly dissected cochlea from P30 rtTA/tet-*Myc*/tet-*NICD* mouse with typical stereocilia in V-shape in OHCs and linear alignment in IHCs. **v–y** 17 days after Dox/ad-*Atoh1* treatment of the cultured adult rtTA/tet-*Myc*/tet-*NICD* cochlea, regenerated HC-like cells (ESPN[+] or ACTB[+]), transdifferentiated either from dividing SC (**w**) or non-dividing SC (**y**), were shown to take up FM1-43FX. Other surrounding non-HC cells (DAPI[+]/ESPN[−]) did not take up FM1-43FX (**w**). ACTB: actin. Scale bars: 2 µm in **r–u**; 10 µm in other figures.

HC-like cells after Dox-induced NOTCH activation, MHY1485 treatment together with ad-*Atoh1* infection than by Dox-induced NOTCH activation and ad-*Atoh1* infection in cultured rtTA/tet-*NICD* adult cochlea (Fig. 7l, m, o), suggesting that MHY1485 and NICD work synergistically to promote HC regeneration. We conclude that the reprogramming effect by *Myc/NICD* in proliferation and HC regeneration is partially mediated by the mTOR signaling pathway.

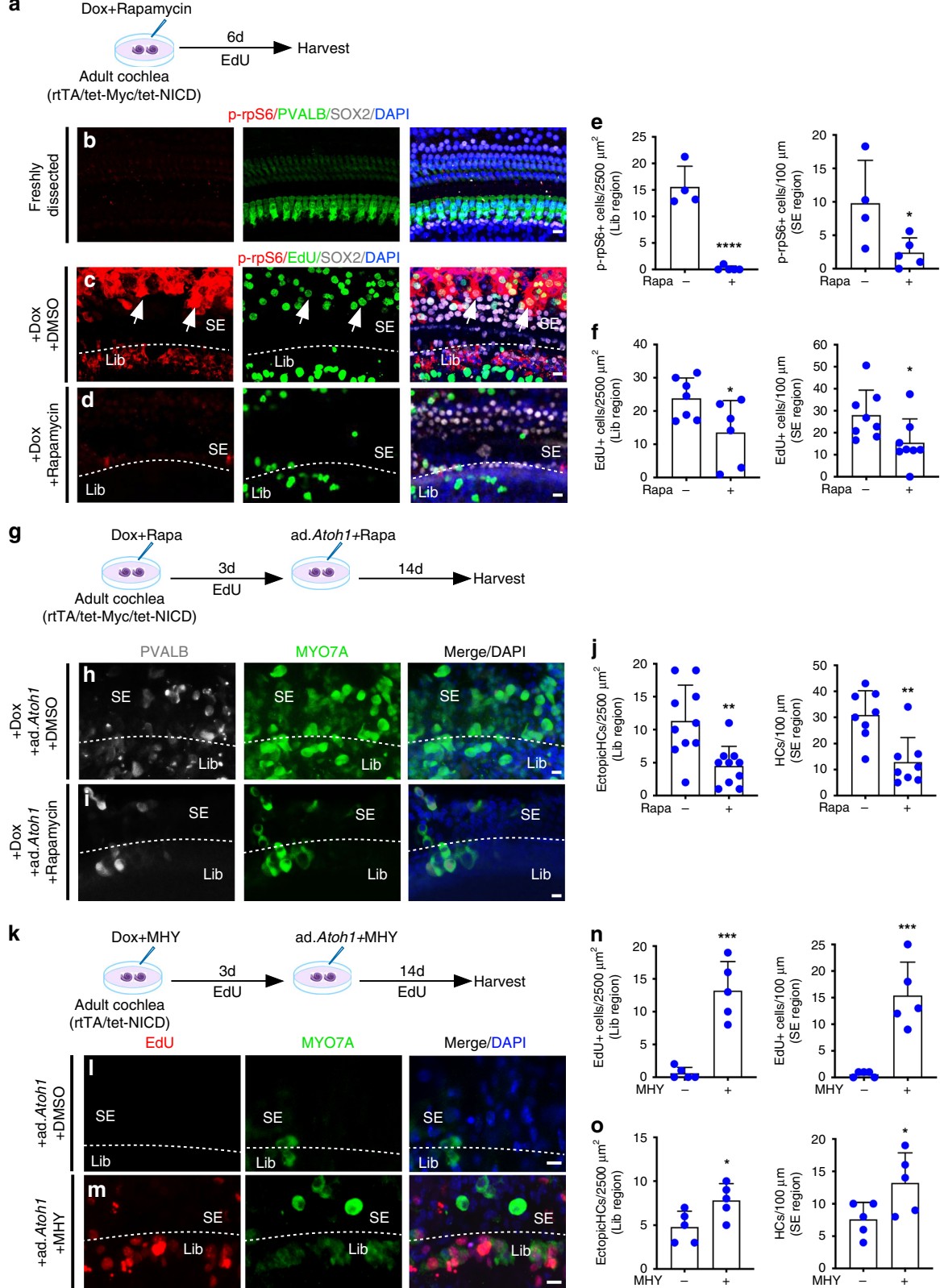

## Discussion

Each of the diverse inner ear cell types, including HCs, SCs, neurons and the cells within the stria vascularis, play an essential role in hearing[45]. The lack of the capacity to proliferate in the damaged adult inner ear has been a major hurdle to regenerate any of the cell types, which ultimately leads to permanent hearing loss. In human newborns, the inner ear is fully mature. Therefore, regeneration of inner ear cells in adult by renewed proliferation and/or transdifferentiation is an essential step towards the development of new treatments for hearing restoration.

By combined activation of *Myc* and *Notch1*, we demonstrate that multiple adult cochlear cell types can be induced to re-enter

**Fig. 7** mTOR is downstream of the MYC/NICD pathway. **a** A diagram illustrating the procedure of Dox and rapamycin treatment on adult rtTA/tet-*Myc*/tet-*NICD* cochlea in vitro. **b** Phospho-rpS6 (p-rpS6), a downstream effector of mTOR signaling, was not detected in freshly dissected adult cochlea. **c** Dox-induced *Myc*/*NICD* activation upregulated p-rpS6 in cultured adult rtTA/tet-*Myc*/tet-*NICD* cochlea. Arrows pointed to the dividing SCs labeled with p-rpS6. **d** 6 days after Dox and rapamycin co-treatment of cultured adult rtTA/tet-*Myc*/tet-*NICD* cochlea, p-rpS6 was greatly reduced and the numbers of EdU$^+$ and p-rpS6$^+$ cells decreased significantly in the sensory epithelial region (SE) and limbus region (Lib) compared to Dox-treated cochlea (**c**). **e**, **f** Quantification of p-rpS6$^+$ and EdU$^+$ cells in the cultured adult rtTA/tet-*Myc*/tet-*NICD* cochleae exposed to DOX, with or without rapamycin treatment. *$p < 0.05$, ****$p < 0.0001$, two-tailed unpaired Student's *t*-test. Error bar, mean ± s.d.; **e**: $n = 5$ for rapamycin-treated groups and $n = 4$ for controls. **f**: Lib region, $n = 6$ for rapamycin-treated group and $n = 7$ for control group; SE region, $n = 8$ for both groups. **g** A diagram illustrating the study of rapamycin treatment on HC induction by *Atoh1*. **h**, **i** The numbers of EdU$^+$ cells and new HC-like cells in the SE and Lib were significantly decreased in the rapamycin-treated adult rtTA/tet-*Myc*/tet-*NICD* cochleae following 3-day Dox treatment and ad-*Atoh1* infection for 14 days. **j** Quantification of total HCs and ectopic HC-like cells (MYO7A$^+$) in the apex turn of cochleae in (**h**, **i**). **$p < 0.01$, two-tailed unpaired Student's *t*-test. Error bar, mean ± s.d.; Lib region, $n = 10$ for both groups; SE region, $n = 8$ for both groups. **k** A diagram illustrating the study of the effect of MHY1485 on proliferation and HC regeneration. **l**, **m** Significantly more EdU$^+$ cells and new HC-like cells in SE and Lib in the apex turn were observed in the MHY1485-treated adult rtTA/tet-*NICD* cochlea following 3-day Dox treatment and ad-*Atoh1* infection for 14 days, compared to DMSO-treated controls. **n**, **o** Quantification of EdU$^+$ cells, MYO7A$^+$ HCs and ectopic HC-like cells in (**l**, **m**). *$p < 0.05$, ***$p < 0.001$, two-tailed unpaired Student's *t*-test. Error bar, mean ± s.d.; $n = 5$ for each group. *n* is the number of biologically independent samples. Scale bars: 10 μm. Source data are provided as a Source Data file.

and complete the cell cycle with dividing cochlear cells maintaining their identities. By transiently modulating MYC/NICD activities, adult cochlear SCs are reprogrammed and can respond to the induction signal by *Atoh1* to transdifferentiate into HC-like cells efficiently in vivo and in vitro. This work forms the basis for our continued study of hearing recovery in adult deaf animal models.

Efforts towards inner ear regeneration have been centered around HCs using a myriad of approaches including *Atoh1* overexpression, inner ear stem cells, induction of proliferation in the cochlea, and, recently, direct conversion of SCs to HCs by multiple transcription factors[11,17–19,46,47]. Because SCs and remaining HCs are already in their proper locations in the cochlea, renewed proliferation and HC regeneration in situ would be ideal for function restoration.

*Myc*, a cell cycle activator and a universal amplifier of expressed genes[30], is not involved in normal inner ear development[48]. *Myc* overexpression induces proliferation in adult mouse utricular SCs[49] and in adult retinal axon regeneration[50]. In zebrafish neuromasts, *myc* is rapidly activated upon HC damage and is required for SC proliferation and HC regeneration[23], supporting the critical role that transient *Myc* activation in renewed proliferation is necessary for HC regeneration. It is significant that terminally differentiated adult cochlear cells are capable of cell cycle re-entry without undergoing drastic de-differentiation: proliferating IHCs regenerate relatively mature IHCs expressing HC marker ESPN (Supplementary Fig. 1i) and IHC marker SLC17A8 (VGLUT3)[51] (Fig. 2n); whereas adult cochlear SC subtypes in cell division express SC markers (Supplementary Fig. 4). The maintenance of cell identity during proliferation is likely due to limited reprogramming, which is supported by up-regulation of inner ear progenitor genes but not stem cell genes (Supplementary Fig. 7f). We have previously shown that Notch inhibition promotes the Wnt pathway, which promotes *Lgr5*$^+$ cochlear progenitor cells to enter cell cycle and regenerate HCs in neonatal mice[52]. Compared to the current study, it is suggested that NOTCH may play a role in inner ear progenitors that is different from those in the adult cochlea in which Notch activation is required for proliferation of multiple SC subtypes, and renewed proliferation does not depend on the presence of progenitor cells.

Our study illustrates a two-stage process in cell cycle re-entry and subsequent HC regeneration. First, SCs are reprogrammed by *Myc*/*NICD* co-activation that is necessary to induce reprogramming and renewed proliferation. Subsequently, *Myc*/*NICD* needs to be down-regulated to allow reprogrammed SCs to respond to the HC induction signals (e.g. *Atoh1* overexpression) to transdifferentiate into HC-like cells. Downregulation of NOTCH is

necessary to HC regeneration likely because NOTCH activates the targets *Hes1* and *Hes5* which are both antagonists of *Atoh1*[35]. For HC regeneration, *Hes1* and *Hes5* need to be silenced, which can be achieved by Notch downregulation and *Atoh1* overexpression. Further, transient *Myc* function is likely important in SC-to-HC transdifferentiation, as it is known that continuous overexpression of *Myc* is detrimental to differentiation[53].

In the rtTA/tet-*Myc*/tet-*NICD* mouse model in vivo, HC-like cells are regenerated from the IHC region almost exclusively by transdifferentiation of non-dividing SCs after reprogramming; whereas the HC-like cells are regenerated from transdifferentiation of dividing and non-dividing cells in the limbus region (Fig. 4d–f; Supplementary Fig. 8a, b). The data strongly supports that *Myc*/*NICD*-mediated reprogramming, not proliferation per se, is a prerequisite to robust SC-to-HC transdifferentiation. HC-like cells can be regenerated from multiple regions within the cochlea including the limbus region and the Claudius cell region, in addition to the sensory epithelial region. Thus, multiple adult cochlear cell types have the potential to become HC-like cells. For hearing restoration, however, it is preferable that regeneration could be confined to the sensory epithelial region so that regenerated HCs may directly contact the tectorial membrane to elicit mechanical transduction. In addition to *Atoh1* overexpression, our study points to an alternative approach of active blockade of Notch function following *Myc*/*NICD* co-activation by interrupting the binding of NOTCH and its partner RBPJ with inhibitors or suppressing *Hes1*/*Hes5* by siRNAs, to regenerate HCs long after HC damage. These transitory approaches may enhance HC maturation and survival in comparison to continuous *Atoh1* overexpression, which is incompatible with HC maturation[54].

An understanding of the mechanism underlying the differences between transient and sustained *Myc*/*NICD* activation should shed light on SC reprogramming, renewed proliferation and improvement in HC regeneration efficiency. The mTOR pathway plays important roles in cell growth, survival and proliferation through regulation of multiple cell processes[55,56]. Up-regulating mTOR activity by pTEN deletion leads to constitutive neurogenesis in brain and regeneration of axons[57,58]. In the mammalian inner ear, the mTOR pathway is involved in proliferation of vestibular SCs that can be suppressed by the mTOR inhibitor rapamycin[59]. The discovery of the mTOR pathway as a major downstream component of *Myc-Notch1* co-activation in the induction of proliferation and HC regeneration efficiency is consistent with the roles of mTOR. As mTOR is not active in normal adult cochlea (Fig. 7b), reactivation of the mTOR pathway together with *Notch1* may provide another route to induce limited reprogramming and proliferation or HC regeneration. Such a combination may allow us to fine-tune the process of

reprogramming related to proliferation and HC regeneration, so the number of cell cycles can be precisely controlled to produce a defined number of HCs and SCs for hearing recovery.

Well documented is the presence of undifferentiated flat epithelial cells instead of SCs in the cochlea of patients suffering from severe hearing loss[60]. The flat epithelium loses responsiveness to *Atoh1* and does not become HCs[61]. Our study points to the possibility of deeper reprogramming, which could be achieved by activation of the otic genes upstream of Notch during development with *Myc* in the flat epithelium so these cells can first become SC-like and then transdifferentiate into HCs.

Regenerated HC-like cells are capable of promoting outgrowth of adult neurites, strongly supporting that new HCs play a dominant role in attracting adult neurites to re-establish HC-neuron connections. Elucidation of the underlying mechanism may lead to strategies for recovery of the lost connections between HCs and neurons in hidden hearing loss[62]. Altogether, our study lays a foundation to study hearing recovery through regeneration of HCs, SCs, and reconnection to neurons in the adult mammalian inner ear.

Reprogramming and proliferation induced by *Myc*/NICD co-activation in the adult cochlea is widespread, encompassing SCs, HCs, and fibrocytes in the stria vascularis, providing a useful tool to regenerate diverse inner ear cell types involved in different types of deafness[63,64]. Given the important roles *Myc* and Notch play in different tissues[65–67], we hope that our study can serve as a model for regeneration of other post-mitotic tissues such as the retina and the central nervous system.

## Methods

**Mouse models**. Rosa-NICD, Rosa-rtTA (rtTA) and tdTomato reporter (tdT) mice were from Jackson Laboratory (Stock# 008159, 006965 and 007914); tet-on-*Myc* mice[68] were from Dr. M. Bishop of the University of California, San Francisco; and tet-on-*NICD* mice[69] were from Dr. D. Melton of Harvard University. For the transgenic mice, the background was mixed C57/129SvJ, with roughly equal numbers of sexes. The wild type mice were C57BL/6 from Charles River Laboratories.

**Viral Injection in vivo**. All adult mice used were between 6 weeks to 8 weeks old, except for those specifically mentioned. For viral injection, cochleostomy was performed on the anesthetized mice by opening the bulla, and adenovirus with a titer of $5 \times 10^{12}$ pfu/ml was injected into the scala media by a pressure-controlled motorized microinjector at the speed of 3 nl/sec. A total of 0.32 μl of adenovirus was injected into each cochlea. For viral mixtures, an equal volume was used for each virus. BrdU (Sigma) or EdU (Santa Cruz Biotech) was injected to the mice i.p. 16 h after viral injection at a concentration of 50 mg/kg daily for 4 or 6 days, with the injected animals harvested at various time points (4, 8, 12, 15 and 35 days). The use of animals was approved by the Massachusetts Eye & Ear Infirmary IACUC, with the treatment of animals following the approved protocols. The viral use was approved by the Harvard COMS committee.

**Induction of *Myc*/NICD in vivo**. To induce *Myc*/NICD in vivo, otic bullas were opened to expose the round window niche. For transient Dox exposure, gelfoam sponge (Pfizer) soaked with Dox in DMSO at the concentration of 50 mg/ml was placed outside the round window niches for 4 days. The gelfoam was surgically removed 2 days before ad-*Atoh1* infection.

**Adult cochlear culture and viral infection in vitro**. Different from the neonatal cochlea culture method, in which the cochleae were disassociated from the bone, adult mouse whole cochleae (6–8 weeks old) were dissected with the bone attached. The bulla was first removed from the skull, with the bulla briefly dipped in 70% ethanol before being placed in ice-cold HBSS. The vestibular region was also removed. Under a dissecting microscope, the middle ear, vessels and the debris were removed from the bulla. The bone covering the apical turn were removed, and round window and oval window membranes were opened to allow media exchange with the cochlea fluids. The ligament portion and Reissner's membrane at each end of the cochlea were also removed to facilitate the access of medium to the sensory epithelial region. The cochleae were maintained in floating culture in DMEM/F12 (Invitrogen) supplemented with N2 and B27 (both from Invitrogen) for 3–22 days. For infection, adenovirus was added to the culture at a titer of $5 \times 10^{10}$ pfu/ml overnight before the replacement with fresh medium. ad-*Atoh1* and ad-*V5* were purchased from the SignaGen Laboratories, Rockville, MD. ad-*NICD*-V5 was a gift

from Dr. Igor Prudovsky, Maine Medical Center Research Institute. ad-*Cre*-GFP and ad-GFP were purchased from the Vector Lab, Baylor College of Medicine, Houston. *Ad-Myc* has been previously described[27]. To label proliferating cells, BrdU and EdU were added at a final concentration of 10 μM for varying time periods.

**Induction of *Myc*/NICD in vitro for HC regeneration**. For in vitro studies, cultured 6-week-old Dox-inducible Rosa-rtTA/tet-*Myc*/tet-*NICD* cochleae were treated with Dox (Sigma, 2 μg/ml final concentration) for 3 days with EdU (10 μM), followed by ad-*Atoh1* ($5 \times 10^{10}$ pfu/ml) infection overnight. The controls were cultured adult cochleae of the same genotype treated with Dox or infected with ad-*Atoh1*, respectively. The culture was replaced with fresh medium for additional 14 days. In the studies of the mTOR pathway, rapamycin (Sigma, 10 μM final concentration) or MHY 1485 (Millipore, 20 μM final concentration) were added to the culture with Dox. The cochleae were harvested and decalcified before immunohistochemistry.

**Lineage tracing**. For in vivo lineage tracing studies, 6-week-old Sox2-CreER/tdT[f/f] mice were injected with tamoxifen (Sigma, 200 mg/kg) daily for three days before injection of ad-*Myc*/ad-*NICD* (0.3 μl per injection) into the cochlea via cochleostomy. BrdU or EdU (50 mg/kg) was injected i.p. daily for 6 days. ad-V5-injected mice were controls. Six to 12 days after ad-*Myc*/ad-*NICD* injection, the mice were sacrificed, and cochleae dissected for processing.

For in vitro lineage tracing studies with triple virus, 6-week-old Sox2-CreER/tdT[f/f] mice were injected with tamoxifen (75 mg/kg) daily for three days before the cochleae were dissected for culture. Triple virus mixture (ad-*Myc*/ad-*NICD*/ad-*Atoh1*) was added to the medium for 16 h at a concentration of $10^9$ pfu/ml per virus. Fresh medium was subsequently used with EdU (10 μM) for culture for additional 14 days before harvest.

**Reporting summary**. Further information on research design is available in the Nature Research Reporting Summary linked to this article.

## Data availability
The source data underlying Fig. 1n, o, 2h–k, q–t, 4g–j, 5e, f, 7e, f, j, n, o, Supplementary Figs. 1d, 2e, f, 4d, g, 5n, and 7e, f are provided as a Source Data file.

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

## Acknowledgements

We are grateful to David Corey for critical reading of the manuscript. We would like to thank Shan Sun, Sofia Chen for help with the figures, Haobing Wang for help with the movies and Corena Loeb for editing the manuscript. The authors acknowledge funding from NIH R01DC006908, R56DC006908 (Z-Y.C.), DOD W81XWH1810331 (Z-Y.C.), Fredrick and Ines Yeatts hair cell regeneration fellowship (Z-Y.C.) and David-Shulsky Foundation (Z-Y.C.), the Major State Basic Research Development Program of China (2011CB504506)(Z.W.), (2011CB504500)(H.L.), Key Laboratory of Hearing Medicine,

National Health and Family Planning Commission, Shanghai, China (Z.W.; H.L.), the National Natural Science Foundation of China (No. 81230019)(H.L.) (No. 81822011, 81771013) (Y.S.), Science and Technology Commission of Shanghai Municipality (17ZR1448600, 18410712400) (Y.S.), NIH R01DC005575, R01DC012115 (X.L.) and NIH R01DC017166 (A.A.I.).

## Author contributions

Y.S, W.L, M.H., and Y.Z.Q. designed and performed experiments, analyzed data, and interpreted results. Y.T, D.S., C.T., and A.A.I. performed experiments and analyzed data. X.L, K.H., and W.Z analyzed data. H.L designed experiments and analyzed data. Z-Y.C. conceived the project, designed experiments, analyzed data, interpreted results and wrote the paper. All authors edited the paper.

## Competing interests

Patent applications based on the work have been filed (Z-Y.C., Y.Q., and W.L.). The remaining authors declare no competing interests.
