## [Peer Review File · Nature Communications]

Reviewers' comments:

Reviewer #1 (Remarks to the Author):

The manuscript from Shu and collaborators represents an extremely important contribution to the field of hair cell regeneration. Given the importance of this work, it is imperative that the presentation of the results be clear and convincing. The authors have largely achieved this, but there are some significant gaps.

That said, I don't want to lose sight of the fact that this manuscript offers several monumental advances in hair-cell regeneration. They show that indeed you can get cell proliferation in adult cochlea, which nobody had been shown before. The MYC/NOTCH1 combo makes sense. Moreover, they show that a transient MYC/NOTCH1 activation leads to cells that are receptive to ATOH1-dependent conversion to new hair cells, a critical first step in defining a hair-cell regeneration paradigm.

Major comments

1. Page 7. The culture system is critically important to this paper, but is described quite perfunctorily. I take it that this method has not been published previously. While it is fantastic to have an adult cochlea culture model, I would like to see more characterization. Clearly the model is not ideal because nearly all the hair cells die. It is important to note therefore that it is a damage model, not a control model. In addition it would be useful to know whether stress pathways or apoptotic pathways are triggered. Certainly the hair cells are not happy in culture, and it is important to know what other effects the culturing has on supporting cells. That said, I do appreciate that the authors generally use living mice for most of the key experiments in the paper, using the culture system to clarify key points.

2. Page 14. You cannot say that the regenerated hair cells are functional. They selectively load with FM1-43, which is consistent with that conclusions, but the Meyers et al. (2003) paper shows that other channels can mediate FM1-43 entry. If the authors want to say that the regenerated hair cells are functional, then they must carry out electrophysiology measurements on these cells and show mechanically activated currents with hair bundle stimulation.

3. Page 14. Likewise, you cannot say that these cells form connections with neurons. For sure, neurons are close by, but for the definitive conclusion, we need to see EM of the synapses and electrophysiology showing functional coupling.

4. There is not enough SEM in the manuscript to convince us that these cells really are on their way to making hair cells. The two magnified images of cilia clusters are nice, but it is critically important to give a broader view of the surface specializations seen. It would also be necessary to see SEM for other conditions as well as the key transient stimulation of proliferation and Atoh1 delivery.

5. Points 2-4 all say the same thing—there is no evidence that bona fide hair cells have been created. You at least need to use the clunky "hair cell-like cells" in referring to these cells. Molecular markers are nice but the key features of generic hair cells are that they transduce mechanical stimuli with specialized hair bundles, and transmit that information to the central nervous system.

6. The figure legends are dense and unreadable. Somehow there needs to be less in the legends. Part of the problem is that the figures are unnecessarily dense themselves. While the split panels for individual colors are useful in some cases, try to reduce the use of them to simply the paper. The poor reader will thank you for this effort.

Other comments

7. Please use official gene, transcript, and protein symbols to refer to the appropriate genes, transcripts, or proteins. For example, instead of c-Myc, please use Myc (italicized for gene and transcript) or MYC (protein). As the authors know, the gene nomenclature is complicated, but is greatly simplified by using the official symbols. Nonstandard symbols (e.g., ESP—the official symbol is ESPN) shouldn't be used. In the list of antibodies in the supplement, make the following changes: AURKB, ESPN, MKI67, NEFH, and SLC17A8. (Note that MYC is correctly referred to here.) I recommend that you use these symbols instead of the common abbreviations throughout the manuscript.

8. Page 4. I would consider cochlear implants to be a medical treatment for hearing loss.

9. Page 4 and beyond. I do not like using "HC" and "SC" to abbreviate hair cell and supporting cell. While those abbreviations make sense to an aficionado, they contribute to making the paper difficult to approach for the naive reader. I would suggest not using the abbreviations and instead writing out "hair cell" and "supporting cell." Yes, it messes with your word count, but you want the manuscript to be readable.

10. Page 4. Why "(SC-HC)"? Is this supposed to be an abbreviation? Rather strange.

11. Page 4. Provide references for supporting cells serving "essential roles in hearing balance."

12. Page 5. What does it mean if studies "failed to reproduce similar results"? It should be "failed to reproduce the essential findings" or "did not find similar results" or something like that.

13. Page 5. NOTCH1, not Notch1. You are referring to the protein, not the gene or transcript.

14. Page 6. "BrdU-labeled cells were detected in postnatal day 7 (P7) cochlear SCs" makes no sense. Either BrdU-labeled cells were detected or BrdU was detected in cochlear supporting cells.

15. Page 8. What do you mean that IHCs were disorganized? In what way?

16. Page 8. Interesting point about size change in hair cells. Was that not seen in supporting cells?

17. Page 8. Is the RT-PCR reaction specifically detecting NICD? If so, say that, as presumably that is overexpressed relative to NOTCH1.

18. Page 9. Why say "Proliferation is primarily the result of Myc and NICD co-activation"? It seems like you can be more forceful in your conclusions.

19. Page 9. Critical mistake in this sentence: "Dividing HCs were detected and were tdT+ (Fig. 3f-i), demonstrating their HC origin." You MUST mean tdT-negative.

20. Page 9. "MYO7A", not "MYo7A."

21. Page 9. What does it mean to say that "[a]bundant SCs were induced to proliferate"? What about the non-abundant SCs? (I am being sarcastic.)

22. Page 9. How can you possibly say that all those supporting cell subtypes can be distinguished in your experiments? I don't buy it; you would need to do lineage tracing experiments.

23. Page 10. It seems bad that the cochleostomy procedure killed all OHCs. Please discuss AND provide quantitative data. (I recognize that the advantage is that you start off in a damage paradigm, but it is not a well characterized paradigm.)

24. Page 11. The following sentence is confusing: "we detected re-expression of inner-ear-progenitor genes but not stem-cell genes (Fig. S6e), including Six1, Eya1, Gata3 and Isl1..." The structure of the sentence implies that those genes are stem cell genes, but they are not, right? Rewrite.

25. Page 11. You don't really mean "In control adult rtTA/tet-Myc/tet-NICD mice injected with ad-Atoh1 for 14 days...", which implies you are injecting ad-Atoh1 over those 14 days. I presume you mean 14 day recovery after injection. Be precise in writing.

26. Page 13. This is a cursory look at specification of OHCs and IHCs. It doesn't add much to the paper.

27. Page 13. Change to: "...examine the presence of the kinocilia, which are only associated..."

28. Page 13. When you refer to "immature concentric stereocilia," how are these defined? SEM can only show surface structure. Do these have appropriate markers of early stereocilia development?

29. Page 14. This sentence is confounding and needs to be rewritten: "The "young" characteristics of transdifferentiation derived new HCs were supported by their long-term survival in culture that is reminiscent of neonatal HCs; whereas all adult cochlear OHCs die rapidly in such conditions."

30. Please label all figures, especially the supplemental figures, with the figure number in the figure itself. There are many figures and it gets confusing if a reviewer makes a mistake and misorders them when assembling them.

31. Fig. 4. There is too much in this figure. It may need to be broken into two figures. The bar charts are basically unreadable because the font is so small. The figures need to be designed so the font sizes are big enough to read and will be at the same size when the figure is in the journal.

32. Fig. 4. How are ectopic hair cells defined? How can you distinguish one from a non-ectopic hair cell?

33. Fig. 4o. What the colors? The label only indicates EdU in red.

34. Fig. 5d. Add the same arrows as in a, c, and e.

35. Fig. 5b-e. The cells pointed out are not EdU positive. I realize that the OHCs and the OHCs are all gone because of the cochleostomy, but why don't the new cells label with EdU?

36. Fig. 4p. Are there really 10x more hair cells? Is that because the hair cells died because of the cochleostomy?

Reviewer #2 (Remarks to the Author):

This study 'Renewed proliferation in adult mouse cochlea and regeneration of hair cells', by Shu et al. is a nicely designed study to investigate the regenerative potential in the cochlea driven by Notch, Atoh1 and c-Myc. They designed a study to transiently co-express c-Myc and NICD with Dox and then inject with Ad-Atoh1. This is likely an important step as continued overexpression of NICD would be inhibitory towards HC formation. The authors also identified a potential downstream signaling pathway, mTOR, that is partially responsible for stimulating proliferation. This a novel approach to stimulating HC regeneration by way of proliferation and will be of interest to those dealing with organ systems that exit the cell cycle during early stages of development. Pending some additional quantification and clarification, I recommend this manuscript for publication.

Major comments:

For all quantifications, do the authors have total cell counts? How many regions did they select for quantification and how did they select base, mid-base, apical region for their quantification?

It is unclear what is the rationale for performing some of the experiments in vitro versus in vivo, when the latter with is feasible with the cochleostomy technique.

Figure 2 and 3: In vivo images are cropped very close to the IHC region only. I would like to assess the OHC region as well. Please show an uncropped image.

How many dividing cells came from existing HCs? i.e. EdU+, tdT-

In Figure 5, the authors show SEM of HC bundles of newly regenerated HCs presenting themselves in concentric circles. Please show an example of bundles of HCs regenerated in vivo.

Minor comments:

Authors should proof read for grammatical errors throughout the manuscript.

The authors state the pH3 is a marker for proliferation cells in Metaphase. This is technically incorrect as it is more of a marker for M phase i.e. all phases of mitosis.

It is not clear to me what 'traditional culture method' implies in Figure S2. Please elucidate in the methods section.

We thank the reviewers for their valuable and thorough comments. Please see our point by point response to the reviewer comments

Reviewer #1:

Major comments:

1. Page 7. The culture system is critically important to this paper, but is described quite perfunctorily. I take it that this method has not been published previously. While it is fantastic to have an adult cochlea culture model, I would like to see more characterization. Clearly the model is not ideal because nearly all the hair cells die. It is important to note therefore that it is a damage model, not a control model. In addition it would be useful to know whether stress pathways or apoptotic pathways are triggered. Certainly the hair cells are not happy in culture, and it is important to know what other effects the culturing has on supporting cells. That said, I do appreciate that the authors generally use living mice for most of the key experiments in the paper, using the culture system to clarify key points.

Response: The adult cochlea explant culture system is a major advance that allowed us to perform studies of induction of proliferation and hair cell regeneration *in vitro*. With the system we showed that while endogenous hair cells die, a majority of supporting cells survive for up to three weeks.

To directly address potential stress/death pathways, we performed new Casp3 labeling experiments using cultured adult cochleae, which showed minimum number of cell death in supporting cells and was consistent with supporting cell survival that facilitates hair cell regeneration (Supplementary Fig. 5).

We further took advantage of the explant culture system to conduct new experiments to determine cell death pathways in the context of activation of MYC/NICD. We showed that with excessive MYC/NICD activities (rtTA/tet-*Myc*/tet-*NICD* cochlea treated with Dox for 14 days or Rosa-NICD cochlea infected with adenovirus to overexpress *Myc/NICD*), heightened apoptosis was induced in supporting cells. In contrast, with transient MYC/NICD activation (rtTA/tet-*Myc*/tet-*NICD* cochlea treated with Dox for 7 days), cell death was minimally induced. This study demonstrates that transient MYC/NICD activation has the essential roles in the promotion of supporting cell survival after reprogramming and their efficient transdifferentiation to HC-like cells. We incorporated the new experiments and data into the manuscript (p10; p14; Supplementary Fig. 5).

In the manuscript, we provided sufficient information for whole adult cochlea explant culture experiment. We have performed comprehensive studies on the establishment of the adult explant

culture system and its broad applications. However, the studies are beyond the scope of the current manuscript and will be reported in a separate manuscript that is in preparation.

2. Page 14. You cannot say that the regenerated hair cells are functional. They selectively load with FM1-43, which is consistent with that conclusion, but the Meyers et al. (2003) paper shows that other channels can mediate FM1-43 entry. If the authors want to say that the regenerated hair cells are functional, then they must carry out electrophysiology measurements on these cells and show mechanically activated currents with hair bundle stimulation.

Response: We agree with the comment that the FM1-43 uptake is not direct evidence of functional hair cells. Currently it is technically challenging to perform electrophysiology study on regenerated hair cells in adult cochlea due to fact the cochlea is encased in the hardest bone inside the body. We thus used one of the accepted criteria for the presence of transduction complex in hair cells by FM1-43 uptake (Geng R, et al., J Neurosci 2013; Kawashima Y et al., J Clin Invest 2011), as long as the incubation time is sufficiently brief (i.e. 15 sec).

3. Page 14. Likewise, you cannot say that these cells form connections with neurons. For sure, neurons are close by, but for the definitive conclusion, we need to see EM of the synapses and electrophysiology showing functional coupling.

Response: We agree with the reviewer that we do not have the definitive results demonstrating the connections between neurons and new HCs. We re-made the movie to better illustrate that adult neurofibers wrapping around the regenerated HC-like cells, which strongly suggests that neurons and HC-like cells form connections (Supplementary Movie 2). We have revised the text to reflect the changes. We currently do not have the means to conduct TEM study, and we could not conduct the electrophysiology study for the same reason as outlined in Q2.

4. There is not enough SEM in the manuscript to convince us that these cells really are on their way to making hair cells. The two magnified images of cilia clusters are nice, but it is critically important to give a broader view of the surface specializations seen. It would also be necessary to see SEM for other conditions as well as the key transient stimulation of proliferation and *Atoh1* delivery.

Response: We have performed new experiments to regenerate HC-like cells using the adult cochlea explant culture system for the SEM study. The study, in addition to the rtTA/tet-Myc/tet-NICD cochleae treated transiently with Dox followed by ad-*Atoh1* infection, also included the controls of rtTA/tet-Myc/tet-NICD cochleae under varying conditions of no treatment, treatment with Dox alone or infection with ad-*Atoh1* alone. In the rtTA/tet-Myc/tet-NICD cochleae treated with Dox followed by ad-*Atoh1* infection, regenerated ectopic HC-like cells with kinocilia were readily detected by SEM. We provided the new SEM images to illustrate these results (Supplementary Fig. 10).

5. Points 2-4 all say the same thing—there is no evidence that bona fide hair cells have been created. You at least need to use the clunky “hair cell-like cells” in referring to these cells. Molecular markers are nice but the key features of generic hair cells are that they transduce mechanical stimuli with specialized hair bundles, and transmit that information to the central nervous system.

Response: We agree with the reviewer that the ultimate proof will require mechanical stimuli of

the hair bundles to produce transduction currents that are transmitted to the CNS. We therefore use the term “hair cell-like cells” to describe regenerated cells in the manuscript.

6. The figure legends are dense and unreadable. Somehow there need to be less in the legends. Part of the problem is that the figures are unnecessarily dense themselves. While the split panels for individual colors are useful in some cases, try to reduce the use of them to simply the paper. The poor reader will thank you for this effort.

Response: We thank the reviewer for the valuable suggestion. We have reorganized the figures with revised figure legends to simplify the overall presentation.

Other comments

7. Please use official gene, transcript, and protein symbols to refer to the appropriate genes, transcripts, or proteins. For example, instead of c-Myc, please use Myc (italicized for gene and transcript) or MYC (protein). As the authors know, the gene nomenclature is complicated, but is greatly simplified by using the official symbols. Nonstandard symbols (e.g., ESP—the official symbol is ESPN) shouldn't be used. In the list of antibodies in the supplement, make the following changes: AURKB, ESPN, MKI67, NEFH, and SLC17A8. (Note that MYC is correctly referred to here.) I recommend that you use these symbols instead of the common abbreviations throughout the manuscript.

Response: We have made the changes to only use the official nomenclature for the symbols.

8. Page 4. I would consider cochlear implants to be a medical treatment for hearing loss.

Response: We have rephrased the sentence to state that there is no pharmacological therapeutics for hearing loss (p4).

9. Page 4 and beyond. I do not like using “HC” and “SC” to abbreviate hair cell and supporting cell. While those abbreviations make sense to an aficionado, they contribute to making the paper difficult to approach for the naïve reader. I would suggest not using the abbreviations and instead writing out “hair cell” and “supporting cell.” Yes, it messes with your word count, but you want the manuscript to be readable.

Response: We agree that the manuscript should be readable for the general public. That being said, the usage of certain abbreviations is necessary to make the paper concise and easier to read. HC and SC abbreviations are widely included in many hearing research papers published in journals such as Nature and Nature Communications. We have made sure that abbreviations are well described upon first use.

10. Page 4. Why “(SC-HC)”?. Is this supposed to be an abbreviation? Rather strange.

Response: That was supposed to be “SC-to-HC”, which has been corrected.

11. Page 4. Provide references for supporting cells serving “essential roles in hearing balance.”

Response: We provide a reference for the statement, ref 10.

12. Page 5. What does it mean if studies “failed to reproduce similar results”? It should be “failed to reproduce the essential findings” or “did not find similar results” or something like that.

Response: We thank the review for pointing out the lack of clarity. We meant to state that the studies following the initial reports on hair cell regeneration/hearing recovery failed to either regenerate hair cells or restore hearing. We changed the sentence to “failed to reproduce the essential findings” in the manuscript (p5).

13. Page 5. NOTCH1, not Notch1. You are referring to the protein, not the gene or transcript.

Response: We are referring to the *Notch1* gene, as in “overexpression of *Notch1*”. We made a change to express it clearly (p5).

14. Page 6. “BrdU-labeled cells were detected in postnatal day 7 (P7) cochlear SCs” makes no sense. Either BrdU-labeled cells were detected or BrdU was detected in cochlear supporting cells.

Response: We changed to the wording as following: BrdU was detected in postnatal day 7 (P7) cochlear SCs (p6).

15. Page 8. What do you mean that IHCs were disorganized? In what way?

Response: We meant that dividing IHCs did not always position themselves next to each other along the length the cochlea. To avoid the confusion however, we removed the description on disorganized IHCs following proliferation.

16. Page 8. Interesting point about size change in hair cells. Was that not seen in supporting cells?

Response: We could clearly detect the volume of HCs because we used ESPN that labels the whole cells. In contrast we primarily used SOX2 to label SC nuclei, thus we are unable to determine the volume of SCs.

17. Page 8. Is the RT-PCR reaction specifically detecting NICD? If so, say that, as presumably that is overexpressed relative to NOTCH1.

Response: The reviewer is correct, the RT-PCR reflected the expression of both NICD and endogenous *Notch1*.

18. Page 9. Why say “Proliferation is primarily the result of Myc and NICD co-activation”? It seems like you can be more forceful in your conclusions.

Response: We agree with the comment and have changed the sentence to “Proliferation is the direct result of Myc and NICD co-activation” (p12).

19. Page 9. Critical mistake in this sentence: “Dividing HCs were detected and were tdT+ (Fig. 3f-i), demonstrating their HC origin.” You MUST mean tdT-negative.

Response: We thank reviewer for pointing out the misstatement. We have changed the description to “dividing HCs were detected and all were found to be tdT⁺ (Fig. 3c), demonstrating” (p10).

20. Page 9. “MYO7A”, not “MYo7A.”

Response: It has been corrected.

21. Page 9. What does it mean to say that “[a]bundant SCs were induced to proliferate”? What about the non-abundant SCs? (I am being sarcastic.)

Response: We rewrote the paragraph to better state the results (p9 & p10).

22. Page 9. How can you possibly say that all those supporting cell subtypes can be distinguished in your experiments? I don’t buy it; you would need to do lineage tracing experiments.

Response: We based the statement largely on the data demonstrating labeling of S100A1 and JAG1, and the position of SCs (e.g. outer pillar cells, Deiters’ cells, Hensen cells and Claudius’ cells) in the cochlea. We agree with the reviewer that we cannot be sure all SC subtypes are induced to divide. We have rephrased the sentence to reflect the point “ Judging by their locations, likely dividing Deiters cells ...” (p39).

23. Page 10. It seems bad that the cochleostomy procedure killed all OHCs. Please discuss AND provide quantitative data. (I recognize that the advantage is that you start off in a damage paradigm, but it is not a well characterized paradigm.)

Response: We have described that the cochleostomy procedure in adults kills a majority of the OHCs around the injection site (see ref 28). As a result, cochleostomy is a damage model. The information has been added (p7).

24. Page 11. The following sentence is confusing: “we detected re-expression of inner-ear-progenitor genes but not stem-cell genes (Fig. S6e), including Six1, Eya1, Gata3 and Isl1...” The structure of the sentence implies that those genes are stem cell genes, but they are not, right? Rewrite.

Response: We re-wrote the sentence (p12).

25. Page 11. You don’t really mean “In control adult rtTA/tet-Myc/tet-NICD mice injected with ad-Atoh1 for 14 days...”, which implies you are injecting ad-Atoh1 over those 14 days. I presume you mean 14 day recovery after injection. Be precise in writing.

Response: We are sorry for the lack of precision in our statement. We re-wrote the sentence as “Two weeks after ad-Atoh1 injection, control adult rtTA/tet-Myc/tet-NICD mice ...” (p13).

26. Page 13. This is a cursory look at specification of OHCs and IHCs. It doesn’t add much to the paper.

Response: In the auditory organ, two types of hair cells, IHCs and OHCs, perform distinct functions. It is important to evaluate in our system if specific auditory HCs can be regenerated

based on molecular marker expression. We indeed detected expression of SLC26A5 (prestin) in some regenerated HC-like cells, suggesting that they may be on a path to becoming OHCs. This is in contrast to a previous study (ref 47) in which the authors could not detect SLC26A5 in their regenerated HCs. Our data is one piece of information that will inform us about the future characterization study of specialized auditory HCs.

27. Page 13. Change to: "...examine the presence of the kinocilia, which are only associated..."

Response: We made the change as suggested (p15).

28. Page 13. When you refer to "immature concentric stereocilia," how are these defined? SEM can only show surface structure. Do these have appropriate markers of early stereocilia development?

Response: The claim is based on the morphology of stereocilia observed in regenerated hair cell-like cells. In contrast to mature OHCs with the V-shaped or IHCs with the aligned stereocilia, the stereocilia in the regenerated HC-like cells resemble those of early developmental immature HCs. We have rewritten the sentence to better describe the observation (p15-16).

29. Page 14. This sentence is confounding and needs to be rewritten: "The "young" characteristics of transdifferentiation derived new HCs were supported by their long-term survival in culture that is reminiscent of neonatal HCs; whereas all adult cochlear OHCs die rapidly in such conditions."

Response: In culture, adult auditory HCs die rapidly whereas young auditory HCs survive for weeks. The regenerated HC-like cells behave like young HCs in which they survive in culture, which is consistent with other properties associated with young HCs including the presence of kinocilia, immature stereocilia and the production of PTPRQ. We re-wrote the sentence "Regenerated HC-like cells had characteristics of "young" HCs and, furthermore, were able to survive in culture for two weeks;". (p16).

30. Please label all figures, especially the supplemental figures, with the figure number in the figure itself. There are many figures and it gets confusing if a reviewer makes a mistake and misorders them when assembling them.

Response: We use the labeling system for the supplementary figures as specified by the journal policy, i.e. Supplementary Fig. 1 etc.

31. Fig. 4. There is too much in this figure. It may need to be broken into two figures. The bar charts are basically unreadable because the font is so small. The figures need to be designed so the font sizes are big enough to read and will be at the same size when the figure is in the journal.

Response: We have re-organized the figures including splitting them into separate figures, moving some figures into the supplementary data, increasing the font size, and using the same scale as much as possible. We believe the figure presentation has been significantly improved.

32. Fig. 4. How are ectopic hair cells defined? How can you distinguish one from a non-ectopic hair cell?

Response: We consider regenerated HC-like cells from SCs as ectopic HCs. We have labeled them consistently throughout the manuscript.

33. Fig. 4o. What the colors? The label only indicates EdU in red.

Response: We re-labeled the figure to make the color consistent (Fig. 5).

34. Fig. 5d. Add the same arrows as in a, c, and e.

Response: We added the arrows.

35. Fig. 5b-e. The cells pointed out are not EdU positive. I realize that the OHCs and the OHCs are all gone because of the cochleostomy, but why don't the new cells label with EdU?

Response: We thank the reviewer for this thoughtful question about an important point. Activation by Myc/NICD reprograms adult cochlea in which some cells will re-enter cell cycle whereas others do not, depending on the degree the cells are reprogrammed. However, HC-like cells can be regenerated from SCs undergoing proliferation, as well as from SCs without proliferation, as long as they are reprogrammed (Fig.5d). Thus, reprogramming but not proliferation is a prerequisite to SC-to-HC transdifferentiation. We discussed the point in the paper (p22).

36. Fig. 4p. Are there really 10x more hair cells? Is that because the hair cells died because of the cochleostomy?

Response: This was referred to the explant culture study (now Fig.5), in which all OHCs and a majority of IHCs died. Comparing to the control samples, close to 10x more HC-like cells were regenerated.

Reviewer #2:

Major comments:

1. For all quantifications, do the authors have total cell counts? How many regions did they select for quantification and how did they select base, mid-base, apical region for their quantification?

Response: For *in vitro* study, we mainly counted cells in the apical region; and for *in vivo* study, we counted cells in the mid-base region. This information is included in the methods.

2. It is unclear what is the rationale for performing some of the experiments *in vitro* versus *in vivo*, when the latter with is feasible with the cochleostomy technique.

Response: In our experience, *in vitro* experiments are more consistent and the results from multiple conditions can be compared efficiently. This is due to the fact that the concentrations of Dox can be altered and maintained in the medium, and *ad-Atoh1* tends to have a relatively even infection pattern in cultured cochlea. For *in vivo* studies, there are greater variations due to the procedures such as cochleostomy and the inability to control certain conditions (e.g. Dox doses). We thus used both systems for different purposes.

Figure 2 and 3: In vivo images are cropped very close to the IHC region only. I would like to assess the OHC region as well. Please show an uncropped image.

Response: We included an un-cropped picture in the manuscript.

How many dividing cells came from existing HCs? i.e. EdU+, tdT-

Response: In case of HC division without Atoh1 induction, all dividing HCs came from existing HCs as shown by the lineage tracing study. However, in the *in vivo* study, the surgical procedure induced HC death. As a result, the percentage of dividing HCs may not be that meaningful as the baseline existing HC number varies depending on the survival rate. Further adenovirus does not infect all HCs and the expression of viral genes cannot be controlled.

We cannot determine in triple mice the percentage of HC-like cells that are derived from existing HCs, as it would require lineaging tracing in triple mice that harbor 5 transgenes, which technically is extremely challenging to perform.

In Figure 5, the authors show SEM of HC bundles of newly regenerated HCs presenting themselves in concentric circles. Please show an example of bundles of HCs regenerated *in vivo*.

Response: We have attempted to produce the SEM for *in vivo* regenerated HCs without success. This could be due to the fact that the time required for bundle formation is considerably longer *in vivo* than *in vitro*.

Minor comments:

Authors should proof read for grammatical errors throughout the manuscript.

Response: We thank the reviewer for the suggestion. We have thoroughly revised the manuscript including proof reading for grammatical errors.

The authors state the pH3 is a marker for proliferation cells in Metaphase. This is technically incorrect as it is more of a marker for M phase i.e. all phases of mitosis.

Response: We agree with the reviewer that this particular anti-pH3 antibody (Ser10) could detect the events in other phases of mitosis. We have rewritten the text to reflect the fact precisely (p8).

It is not clear to me what 'traditional culture method' implies in Figure S2. Please elucidate in the methods section.

Response: Traditional culture method refers to the way neonatal cochlea is cultured, in which the sensory epithelia are peeled off from the rest of the cochlea for culture. This procedure, while successful in the culture of neonatal cochlear sensory epithelium, results in complete degeneration of adult cochlea including HCs and SCs (Supplementary Fig. 2c). Thus, we developed the adult cochlea explant culture system for our study. The explant culture system will have broader utility for inner ear studies *in vitro*. We have elaborated on this in the Methods.

REVIEWERS' COMMENTS:

Reviewer #1 (Remarks to the Author):

The resubmission of the manuscript from Shu et al. is improved over the original submission, although it remains a challenging manuscript to wade through. That said, the authors have some very important observations here—in particular, their demonstration of enhancement of proliferation of adult supporting cells and conversion of some of those cells to hair cells. These steps are the holy grail of hair cell regeneration, and while the authors have not demonstrated a viable pathway for a hair cell regeneration treatment, that is not their goal. It is highly significant that they did uncover key steps that should be useful in the future in designing bona fide hair cell regeneration therapeutic approaches.

However, I am more than annoyed that this sentence remains in the abstract: “These regenerated hair cells have functional transduction channels and are likely to form connections with adult spiral ganglion neurons...” (1) As was commented in my original review, AND as agreed to by the authors in their rebuttal letter, you CAN'T say these are hair cells. They are hair-cell-like cells. Sorry. It's rather astonishing that this remains. (2) Likewise, you CAN'T say that these cells have functional transduction channels. Period. What if they upregulated some other large-conductance channel that admitted FM1-43? The authors' argument that the short incubation time shows that this is transduction is simply wrong. The short incubation time means that loading via channels (of any sort that passed FM1-43) will be favored over loading via endocytosis. Do not oversell the story.

Likewise, the SEM is a little better but seeing just a few cells is not ideal. Clearly these cells have some features reminiscent of hair cells, but they are not normal hair cells.

Finally, while the adult cochlea culture system is a significant advance, I remain very troubled that all of the hair cells die in it.

Other than the comments above, I do feel that the authors adequately addressed my original critique. In summary, I find that the highly significant results in this manuscript counterbalance the weaknesses, albeit just barely.

Reviewer #2 (Remarks to the Author):

The authors of this study have addressed all the reviewers' comments. Furthermore, they have shown that prolonged Myc/NICD activation leads to increase CASP3 activation, but upon transient Myc activation there was an improved survival rate. I recommend this manuscript for publication.

Comment: It would seem to me that the paragraph on page 14 addressing the improved survival of transient Myc/ NICD activated cochleas, might be better following the CASP3 experiment on page 10.

Reviewer 1.

Comment: (1) As was commented in my original review, AND as agreed to by the authors in their rebuttal letter, you CAN'T say these are hair cells. They are hair-cell-like cells. Sorry. It's rather astonishing that this remains. (2) Likewise, you CAN'T say that these cells have functional transduction channels. Period.

Reply: We have changed the descriptions in the abstract and throughout the manuscript. We included the sentence " These regenerated hair cell-like cells take up the styryl dye FM1-43 and are likely to form connections with adult spiral ganglion neurons, supporting that *Myc* and *Notch1* co-activation is sufficient to reprogram fully mature supporting cells to proliferate and regenerate hair cell-like cells in adult mammalian auditory organs."

Comment: Likewise, the SEM is a little better but seeing just a few cells is not ideal. Clearly these cells have some features reminiscent of hair cells, but they are not normal hair cells.

Reply: We have included more SEMs to show immature stereocilia of HC-like cells (Supplementary Fig. 10f-i). We have shown more SEM images in the manuscript than most publications on hair cell regeneration. Our data have been consistent in that HC-like cells derived from SC transdifferentiation resemble young HCs with immature stereocilia. Future work will include the promotion of maturation of regenerated HC-like cells.

Comment: Finally, while the adult cochlea culture system is a significant advance, I remain very troubled that all of the hair cells die in it.

Reply: The adult cochlea whole culture system, despite HC death, is very valuable to the study of HC regeneration. Because of death of existing HCs, it makes it easier to identify regenerated HC-like cells. Further, regenerated HC-like cells survive the culture system, similar to neonatal HCs, which further supports that the regenerated HC-like cells resemble young HCs.

Reviewer 2

Comment: It would seem to me that the paragraph on page 14 addressing the improved survival of transient *Myc*/ *NICD* activated cochleae, might be better following the *CASP3* experiment on page 10.

Reply: We considered the suggestion seriously. However, if we describe all the *CASP3* data on p10, it would involve the introduction of the creation of the reversible *Myc*/*NICD* mouse model without a compelling reason, as we built the model to primarily control *MYC*/*NICD* activities reversibly. We used a sentence "While cell death is consistent with high and sustained *MYC* activity due to adenovirus-mediated overexpression, our subsequent experiment showed that under transient *Myc/Notch* activation, proliferating cells can survive", to let readers know more data are coming to support our view.